# Tropical peatland carbon storage linked to global latitudinal trends in peat recalcitrance

Suzanne B. Hodgkins [1,2], Curtis J. Richardson[3], René Dommain[4,5], Hongjun Wang [3], Paul H. Glaser[6], Brittany Verbeke[7], B. Rose Winkler[7], Alexander R. Cobb[8], Virginia I. Rich[2], Malak Missilmani[9], Neal Flanagan[3], Mengchi Ho[3], Alison M. Hoyt[10], Charles F. Harvey[11], S. Rose Vining[12], Moira A. Hough[13], Tim R. Moore[14], Pierre J. H. Richard[15], Florentino B. De La Cruz [16], Joumana Toufaily[9], Rasha Hamdan[9], William T. Cooper[1] & Jeffrey P. Chanton[7]

Peatlands represent large terrestrial carbon banks. Given that most peat accumulates in boreal regions, where low temperatures and water saturation preserve organic matter, the existence of peat in (sub)tropical regions remains enigmatic. Here we examined peat and plant chemistry across a latitudinal transect from the Arctic to the tropics. Near-surface low-latitude peat has lower carbohydrate and greater aromatic content than near-surface high-latitude peat, creating a reduced oxidation state and resulting recalcitrance. This recalcitrance allows peat to persist in the (sub)tropics despite warm temperatures. Because we observed similar declines in carbohydrate content with depth in high-latitude peat, our data explain recent field-scale deep peat warming experiments in which catotelm (deeper) peat remained stable despite temperature increases up to 9 °C. We suggest that high-latitude deep peat reservoirs may be stabilized in the face of climate change by their ultimately lower carbohydrate and higher aromatic composition, similar to tropical peats.

[1] Department of Chemistry and Biochemistry, Florida State University, Tallahassee, FL 32306, USA. [2] Department of Microbiology, The Ohio State University, Columbus, OH 43210, USA. [3] Duke University Wetland Center, Nicholas School of the Environment, Durham, NC 27708, USA. [4] Institute of Earth and Environmental Science, University of Potsdam, 14476 Potsdam, Germany. [5] Department of Anthropology, Smithsonian Institution, National Museum of Natural History, Washington, DC 20013, USA. [6] Department of Earth Sciences, University of Minnesota, Minneapolis, MN 55455, USA. [7] Department of Earth, Ocean, and Atmospheric Science, Florida State University, Tallahassee, FL 32306, USA. [8] Center for Environmental Sensing and Modeling, Singapore-MIT Alliance for Research and Technology, Singapore 138602, Singapore. [9] Laboratory of Materials, Catalysis, Environment and Analytical Methods (MCEMA-CHAMSI), EDST and Faculty of Sciences l, Lebanese University, Campus Rafic Hariri, Beirut, Lebanon. [10] Max Planck Institute for Biogeochemistry, 07701 Jena, Germany. [11] Department of Civil and Environmental Engineering, Massachusetts Institute of Technology, Cambridge, MA 02139, USA. [12] Department of Soil, Water and Environmental Science, University of Arizona, Tucson, AZ 85716, USA. [13] Department of Ecology and Evolutionary Biology, University of Arizona, Tucson, AZ 85716, USA. [14] Department of Geography, McGill University, Montreal, QC H3A 0B9, Canada. [15] Département de Géographie, Université de Montréal, Montréal, QC H2V 2B8, Canada. [16] Department of Civil, Construction, and Environmental Engineering, North Carolina State University, Raleigh, NC 27695, USA. These authors contributed equally: Curtis J. Richardson, René Dommain. Correspondence and requests for materials should be addressed to S.B.H. (email: suzanne.b.hodgkins@gmail.com) or to J.P.C. (email: jchanton@fsu.edu)

Peatlands are a major global carbon reservoir (528–600 Pg), with a significant portion of this carbon mass (10–30%) in tropical peatlands[1–4]. Peat accumulation occurs when net primary productivity exceeds the rate of carbon loss via fires and decomposition, which is inhibited at high latitudes by anaerobic conditions[5] and cold temperatures[6]. The existence of large peat deposits at low latitudes, where year-round warm temperatures would be expected to drive higher microbial decomposition rates[7,8], is thus surprising. Several hypotheses have been proposed to explain the accumulation of peat in these environments, such as higher primary productivity close to the equator[9] that may allow faster litter deposition, as well as physical and chemical peat characteristics that may slow decomposition rates. For example, peat in tropical peat swamp forests is largely composed of coarse woody material from fallen trees, branches, and dead roots[10]. This material may be protected from decomposition by its low surface-area-to-volume ratio and high lignin content[11,12], which has been hypothesized to severely limit its anaerobic decomposition[13,14]. Low-latitude peat decomposition may also be slowed by other chemical processes, including release of decomposition-inhibiting phenolics from shrubs in unsaturated shrub peatlands[15] and high organic matter recalcitrance following initial rapid decay of plant litter[16,17]. These effects can be sufficient to preserve peat even in partially unsaturated conditions[15]. However, their potential to preserve high-latitude peat as the climate warms and as woody species expand remains uncertain.

Here we examined the role of peat and parent plant chemistry, in particular the relative abundances of carbohydrates (i.e., O-alkyl C or polysaccharides) and aromatics, in driving peat formation and preservation along a latitudinal transect of major peatland regions from the Arctic to the tropics (Table 1; Fig. 1). Relative abundances of carbohydrates and aromatics are indicators of organic matter reactivity, with lower carbohydrate and higher aromatic content indicating greater humification and/or recalcitrance[18–21]. In this study, we used a newly developed approach for Fourier transform infrared spectroscopy (FTIR) analysis (see Methods), which is based on area-normalized peak heights calibrated to wet chemistry analyses in a set of standard materials[22], to estimate carbohydrate and aromatic content in peat from high-latitude, mid-latitude, and low-latitude field sites. Because peat chemical composition is strongly affected by parent vegetation in addition to humification[18], we also analyzed selected plant samples to distinguish the effects of humification from those of source plant material. The sites along the latitudinal transect (Table 1; Fig. 1) included a permafrost plateau in Stordalen Mire, subarctic Sweden (68°N; Stordalen: CPP); boreal bogs and fens in northern Minnesota (MN) (47–48°N; MN Bogs: Zim Bog, RL-II Bog, and S1 Bog; MN Fens: Bog Lake Fen and RL-II Fen); a boreal bog near Ottawa, Canada (45°N; Mer Bleue: MB-775 and MB-930); temperate pocosin bogs in North Carolina (NC) with frequent low-intensity fires (35°N; NC Pocosin: DNL and DNL deep); subtropical peat marshes in the Loxahatchee National Wildlife Refuge, northern Everglades (26°N; Loxahatchee: Lox3 and Lox8); and tropical peat swamp forest sites in the Ulu Mendaram Conservation Area in Brunei Darussalam, northwest Borneo (4°N; Mendaram: MDM-III and MDM11-2A).

Our results show that near-surface (sub)tropical peat has lower carbohydrate and greater aromatic content than near-surface Arctic and boreal peat, making (sub)tropical peat more chemically recalcitrant. We propose two main drivers of these trends: first, plants in warmer climates contribute litter that is higher in recalcitrant aromatics and lower in carbohydrates compared to plants in colder climates. Second, more extensive humification in warm climates selectively removes labile carbohydrates and concentrates aromatics, causing a negative feedback to further decomposition. We propose that although anaerobic conditions

are key to peat formation across all climates, other drivers differ between climatic zones and peat depths, with cold temperatures a key factor at high latitudes and more recalcitrant organic matter a key factor at low latitudes and deeper depths.

## Results and Discussion

**Differences in peat preservation mechanisms with latitude.** In this study, we have focused on two solid-phase organic matter components that have been shown to drive peat decomposition: carbohydrates that are the most labile solid-phase component[20], and aromatics that inhibit anaerobic decomposition[14,23]. These components produce distinct peaks in the FTIR spectra (Supplementary Fig. 1; Supplementary Table 1). Based on the techniques used to calibrate these FTIR peaks (see Methods; ref. [22]; Supplementary Fig. 2), carbohydrates consist of acid-hydrolysable polysaccharides, whereas aromatics consist of lignin and other unsaturated acid-insoluble material such as tannins and humic substances. While other components such as aliphatics have been shown to correlate with peat humification[20], these components have not been identified as active in the humification process[24] (unlike carbohydrates that are preferentially lost[20] and aromatics that can actively inhibit decomposition[14,23]), but most likely become concentrated as labile components degrade.

Our results clearly show lower carbohydrate and greater aromatic content in temperate to tropical sites compared with Arctic and boreal sites (Fig. 2). Aliphatic content was slightly higher in temperate to tropical sites, but this difference was much less pronounced (Supplementary Fig. 3). On average, surface peat (<50 cm) north of 45°N had higher carbohydrate than aromatic content, whereas surface peat south of 45°N had lower carbohydrate than aromatic content (Fig. 3a, b; Supplementary Fig. 4). Linear regression analysis (Fig. 3) of surface peat carbohydrate and aromatic contents vs. latitude and mean annual temperature (Supplementary Table 2) showed that these trends were significant. The overall highest aromatic concentration was found in the equatorial Mendaram peatland (Fig. 3). This result is consistent with previous FTIR and lignin phenol analyses at this site[11,12], which showed very high lignin content and smaller carbohydrate peaks than our northern sites.

The latitudinal trends in carbohydrate and aromatic content were also visible via principal components analysis (PCA) of the entire FTIR spectra (Fig. 4), both with and without peat-forming vegetation included. In both PCAs, the loadings of PC1 were most negative in the peak at ~1030 cm$^{-1}$ (used to estimate % carbohydrates) and most positive in the peaks at ~1500 and ~1600 cm$^{-1}$ (used to estimate % aromatics) (Fig. 4a, c). Latitude and temperature varied mainly along PC1, with sites south of 45° N having higher PC1 scores (Fig. 4b, d). These results indicate that global latitudinal trends in peat FTIR spectra are dominated by a decrease in carbohydrates and increase in aromatics toward the tropics.

Although anaerobic conditions are a key factor allowing peat formation across a range of latitudes, the significant trends observed in peat chemistry with latitude and temperature (Fig. 3) indicate climatically driven influences on peat formation. At high latitudes, low temperatures and seasonally frozen soils favor peat accumulation by slowing decomposition. In the case of *Sphagnum* bogs, additional peat preservation mechanisms may include the low N content[25,26], high acidity[25], and high abundance of the decomposition-inhibiting carbohydrate sphagnan[27,28]. With this exception, carbohydrates are more reactive than aromatics[17–21,29] because their greater carbon oxidation state increases thermodynamic energy yields for decomposition[30]. Our results thus demonstrate that at low latitudes, the peat's low carbohydrate and high aromatic content (Figs. 2 and 3) leads to high recalcitrance,

**Table 1 Characteristics and locations of sites along the latitudinal transect**

| Region | Site/core | Peatland type and water table (WT) depth | Dominant vegetation | Climate | Latitude and longitude | Additional location information | References |
|---|---|---|---|---|---|---|---|
| Stordalen | CPP | Permafrost plateau (dry; active layer ~60 cm thick) | lichens, shrubs, *Eriophorum vaginatum* | subarctic | 68.3531°N, 19.0473°E | northern Sweden | 49–51,53,56,63,64,[a] |
| Minnesota: MN Bogs | Zim Bog | bog (WT −19 to −33 cm) | *Sphagnum* spp. | boreal | 47.1791°N, 92.7146°W | n/a | 66 |
| Minnesota: MN Bogs | RL-II Bog | bog (WT −5 to −20 cm) | *Sphagnum* spp. | boreal | 48.2547°N, 94.6976°W | Glacial Lake Agassiz peatlands (GLAP) | 31,61,62,67,88 |
| Minnesota: MN Bogs | S1 Bog | bog (WT 0 to −10 cm) | *Sphagnum* spp. | boreal | 47.5063°N, 93.4527°W | Marcell Experimental Forest; plot T3F in the SPRUCE experiment | 21 |
| Minnesota: MN Fens | Bog Lake Fen | poor fen (WT ~−7 cm) | *Sphagnum* spp. and sedges | boreal | 47.5051°N, 93.4890°W | Marcell Experimental Forest | 66 |
| Minnesota: MN Fens | RL-II Fen | rich fen (inundated) | sedges | boreal | 48.2897°N, 94.7083°W | GLAP | 31,61,62,67,88 |
| Mer Bleue | MB-775 | bog (WT −30 to −40 cm) | *Sphagnum* spp. | boreal | 45.4088°N, 75.5182°W | between bog center and margin | 69[a] |
| Mer Bleue | MB-930 | bog (WT −30 to −40 cm) | *Sphagnum* spp. | boreal | 45.4110°N, 75.5171°W | near center of bog | 69 |
| NC Pocosin | DNL | pocosin (WT ~−30 cm; burned 30 years prior to sampling) | shrubs | temperate, subtropical | 35.6905°N, 76.5282°W | Pocosin Lakes National Wildlife Refuge | 15[a] |
| NC Pocosin | DNL deep | pocosin (WT ~−30 cm; burned 30 years prior to sampling) | shrubs | temperate, subtropical | 35.6904°N, 76.5283°W | Pocosin Lakes National Wildlife Refuge | 15[a] |
| Loxahatchee | Lox3 | peat marsh (inundated; WT +50 to +100 cm) | *Cladium jamaicense* | subtropical | 26.597°N, 80.357°W | Loxahatchee National Wildlife Refuge | 47,48,[a] |
| Loxahatchee | Lox8 | peat marsh (inundated; WT +50 to +100 cm) | *Cladium jamaicense* | subtropical | 26.520°N, 80.335°W | Loxahatchee National Wildlife Refuge | 47,48,[a] |
| Mendaram | MDM11-2A | forested peat dome (WT +20 to −20 cm) | large trees (*Shorea albida*) | tropical | 4.3727°N, 114.3550°E | Ulu Mendaram Conservation Area, Brunei | 10 |
| Mendaram | MDM-III | forested peat dome (WT +20 to −20 cm) | large trees (*Shorea albida*) | tropical | 4.3702°N, 114.3542°E | Ulu Mendaram Conservation Area, Brunei | 11,45 |

[a]Reference describes peatlands in the general area, but does not mention this specific coring site

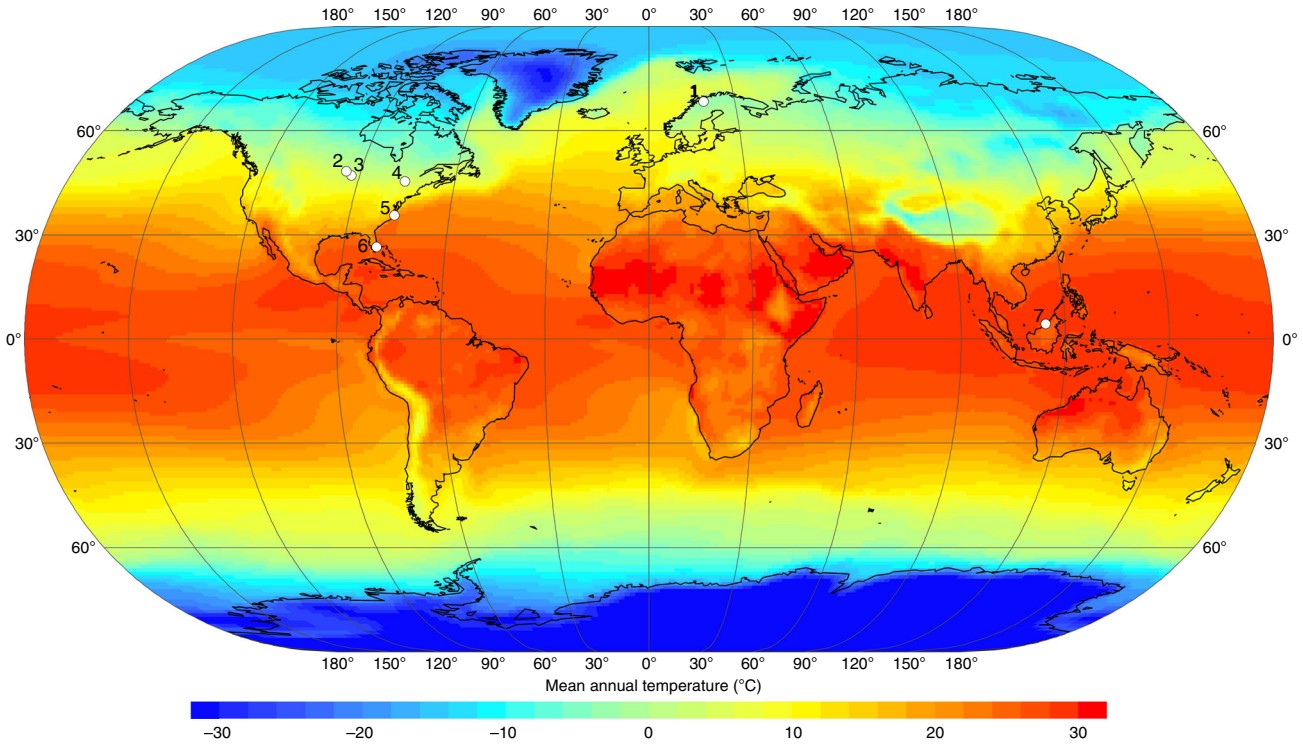

**Fig. 1** Locations of study sites along the global temperature gradient. Sites are shown as numbered white dots: (1) Stordalen (Sweden); (2) RL-II Bog and RL-II Fen (Minnesota, USA); (3) Zim Bog, S1 Bog, and Bog Lake Fen (Minnesota, USA); (4) Mer Bleue (Ontario, Canada); (5) NC Pocosin (North Carolina, USA); (6) Loxahatchee (Florida, USA); (7) Mendaram (Borneo, Brunei). The map shows global mean annual surface temperature in degrees Celsius (°C) (ref. [87])

allowing peat to avoid mineralization and persist in (sub)tropical climates despite warmer temperatures[15,16].

**Drivers of peat chemistry in warm climates.** The trends in peat chemistry with latitude—specifically, the lower carbohydrate and higher aromatic content in tropical and subtropical peatlands—are most pronounced at the surface, whereas northern peat at deeper depths acquires a chemistry more similar to low-latitude peat (Figs. 2 and 4; Supplementary Fig. 5). Moreover, the PCA of FTIR spectra for peat and peat-forming vegetation shows similar

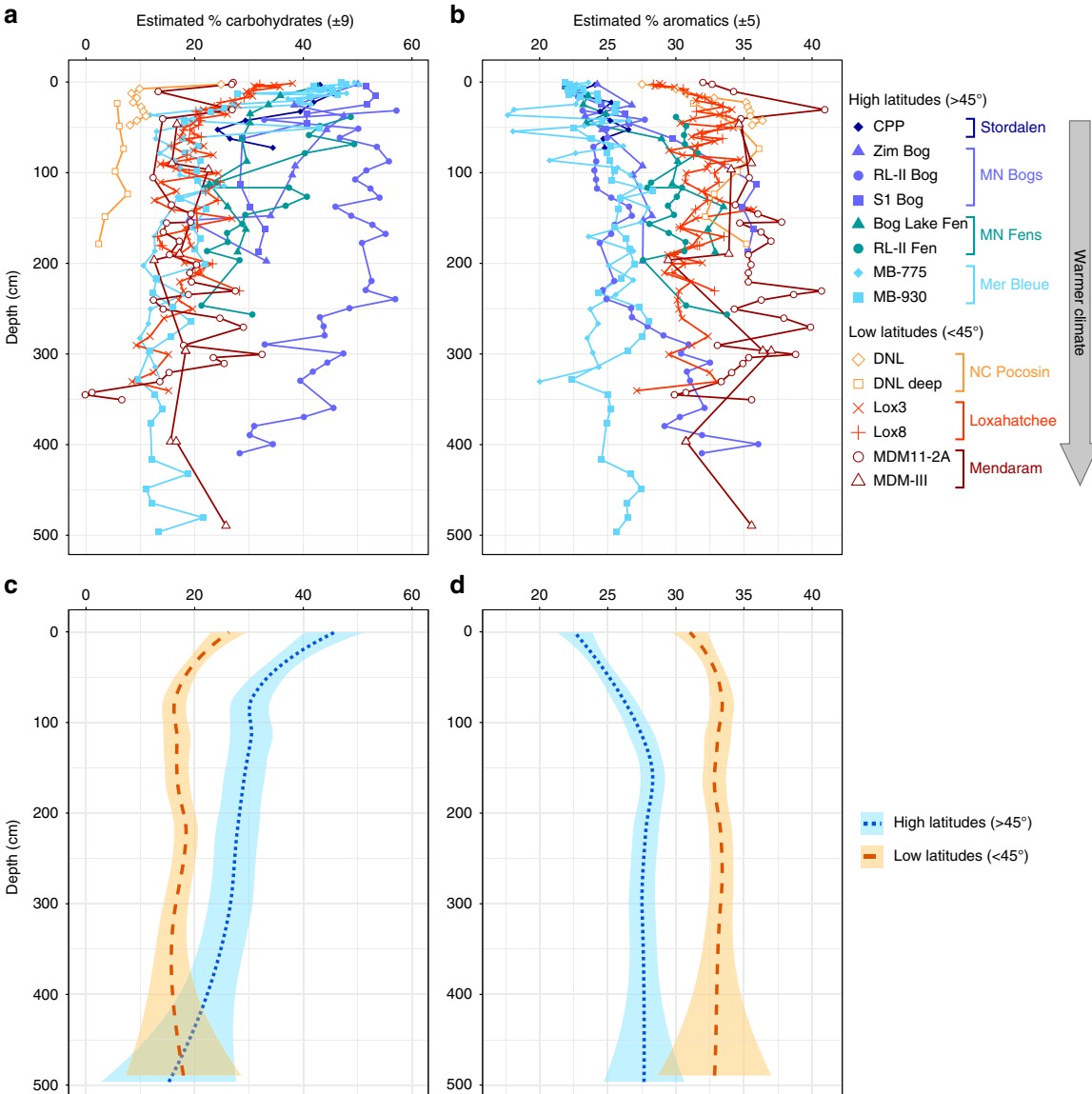

**Fig. 2** Variations in peat chemistry depth profiles across the latitudinal transect. **a**, **b** Estimated weight percentages of **a** carbohydrates and **b** aromatics in individual samples, determined based on Fourier transform infrared spectroscopy (FTIR) peak heights (~1030 cm$^{-1}$ for carbohydrates, and the sum of ~1510 and ~1630 cm$^{-1}$ for aromatics) calibrated to wet chemistry measurements (see Methods). Errors listed in the x-axis for each measurement are the standard errors of the y estimates for the calibrations shown in Supplementary Fig. 2. These depth profiles are also shown separated by peatland category in Supplementary Fig. 4. **c**, **d** General trends for high-latitude and low-latitude peatlands illustrated with locally weighted polynomial regression (LOESS) smooth curves and shaded 95% confidence intervals (LOESS parameters: degree = 2, α = 0.75) for **c** carbohydrates and **d** aromatics, shown for individual cores in **a** and **b**, respectively

latitudinal variations in both categories, although the variations in vegetation are less pronounced, with vegetation from a range of latitudes clustering with high-latitude peat along PC1 (Fig. 4d). These patterns suggest two mechanisms to explain the global trend in peatland organic matter chemistry: (1) the initial chemical quality of the peat-forming plant material (carbohydrate and aromatic content) is changing along the latitudinal transect, such that plant litter and the resulting peat are more recalcitrant at low latitudes, or (2) there is a direct temperature control on the initial rate of labile carbon loss in peatlands, such that surface (sub)tropical peat is already well decomposed, whereas surface northern peat is poorly decomposed and instead degrades more slowly down the profile. In addition, a combination of both mechanisms may have a role in creating this latitudinal trend in peat chemistry.

First, the chemical composition of plant inputs appears to contribute fundamentally to peat recalcitrance. The source vegetation responsible for peat formation varies with latitude, with non-woody *Sphagnum* and sedges dominant within a broad range of colder climates[31,32], and woody trees and shrubs (or less commonly *Cladium* and other sedges if non-forested) dominant within a broad range of warmer climates[2,4,10,33–35]. These plant communities exhibit differences in chemical composition that mirror those seen in the peat, as indicated by comparison of carbohydrate and aromatic content within the peat and the dominant peat-forming plants from different latitudinal zones (Fig. 5; Supplementary Fig. 6; Supplementary Table 3). Based on unpaired two-tailed t tests, plants from low-latitude sites (NC Pocosin, Loxahatchee, and Mendaram) had significantly lower carbohydrate content ($t(37) = 3.412$, $p = 0.002$) and greater

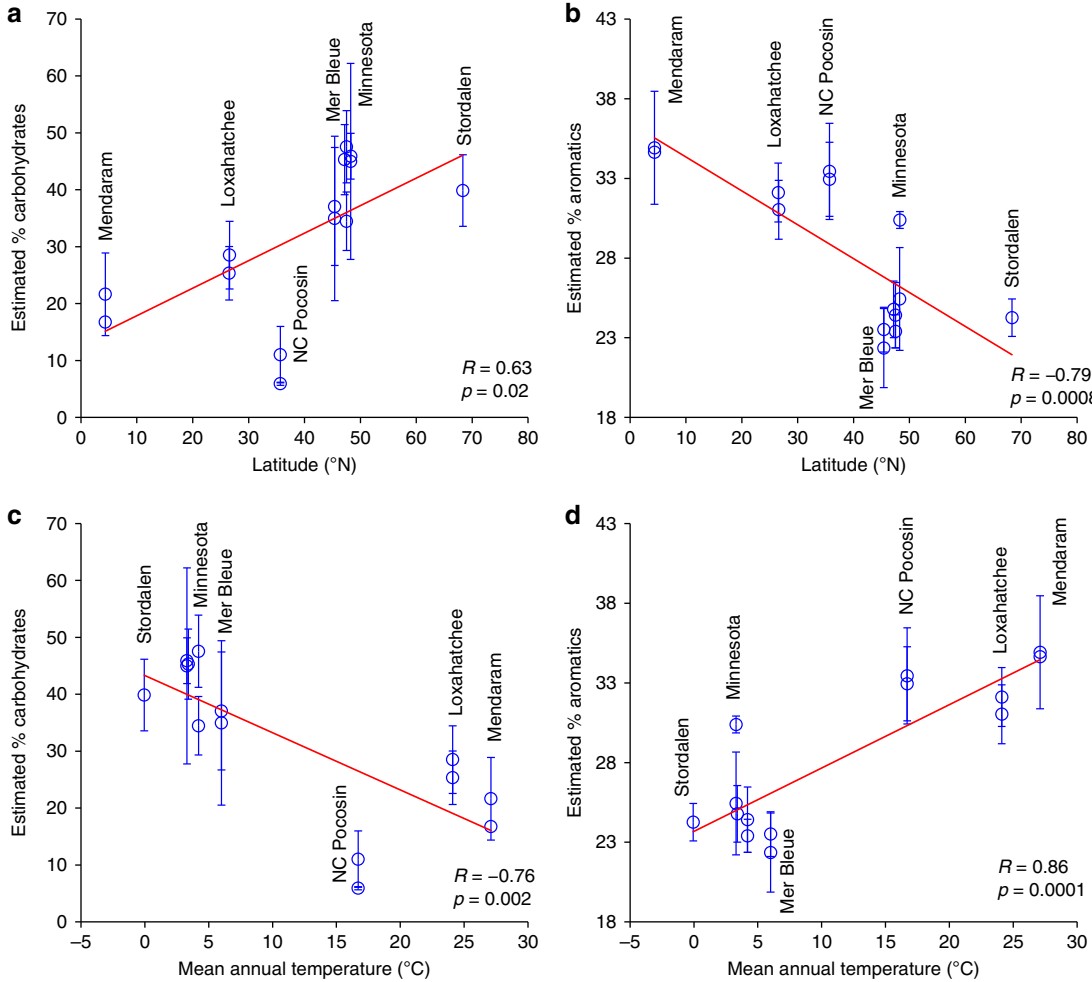

**Fig. 3** Correlations of estimated carbohydrate and aromatic contents with latitude and mean annual temperature. **a** Carbohydrates vs. latitude; **b** aromatics vs. latitude; **c** carbohydrates vs. temperature; **d** aromatics vs. temperature. Each point represents the average ± one standard deviation (SD) of core sections within the top 50 cm of each core (Supplementary Table 2)

aromatic content ($t(37) = 2.652$, $p = 0.01$) than plants from high-latitude sites (Stordalen and Minnesota). In the Mendaram site, lignin from *Shorea albida* wood[11,12] may contribute to the high aromatic content of the peat (Figs. 2b, d and 3b, d). This high lignin content may help to explain the very low anaerobic decomposition and $CH_4$ production rates previously reported in Southeast Asian peatlands[13,14]. In the Loxahatchee, despite the relative sparseness of woody plants, lignin is still abundant in the roots and shoots of *Cladium jamaicense*, which are strengthened by girders (i.e., bundles of sclerenchyma cells)[36]. The abundant aromatics in NC Pocosin peat may include shrub-derived lignin, as well as phenolics that further inhibit decomposition[15]. This latitudinal pattern of increasing aromatics in peatland plants towards the equator may reflect the increase in plant defenses against rising herbivory from high to low latitudes[37]. Especially in tropical forests, strong selective pressures caused by the large diversity of herbivores, particularly insects, has produced an immense variety of chemical plant defense mechanisms[38,39]. The evolution of these defense traits may have given rise to the side effect of inhibiting decomposition and causing peat accumulation[38,40,41]. The low carbohydrate content of (sub)tropical peat, like the high aromatic content, is also influenced by source vegetation. This is particularly true in the NC Pocosin and Mendaram sites, where the dominant plant material had low carbohydrate contents (Fig. 5a). In addition to the high levels of plant-derived aromatics, this low concentration of labile

carbohydrates presumably also contributes to peat recalcitrance at these sites.

Second, higher mean annual temperatures may favor greater humification of tropical and subtropical peat, with the selective removal of reactive carbohydrates and accumulation of aromatics leading to a highly recalcitrant residual peat[17,18,21,29]. Once most of the labile carbohydrates have been removed or transformed, these humification-induced chemical changes act as a negative feedback for further decomposition, preventing complete mineralization to $CO_2$ and $CH_4$[16,17]. Despite carbon loss during humification, the higher primary productivity at low latitudes[9] can still allow the accumulation of thick humified peat deposits[3]. Humification can occur not only via slow transformation within the peat column[24] (detectable as a relative loss of carbohydrates and gain in aromatics with depth: Figs. 2 and 4; Supplementary Fig. 4; Supplementary Fig. 7), but also more rapidly via humification of plant litter at the peat surface. Indications of more extensive humification of litter in warmer climates can be found through comparisons (Fig. 5; Supplementary Fig. 6) of the chemistry of near-surface peat (upper 50 cm; Supplementary Table 2) with that of the plants from which it is derived (Supplementary Table 3). Carbohydrate content was significantly greater in the plants compared to that of the peat in several sites (Fig. 5). However, these differences were greatest and most significant in low-latitude sites (Figs. 4d and 5a), suggesting more rapid loss of carbohydrates following plant inputs in warm

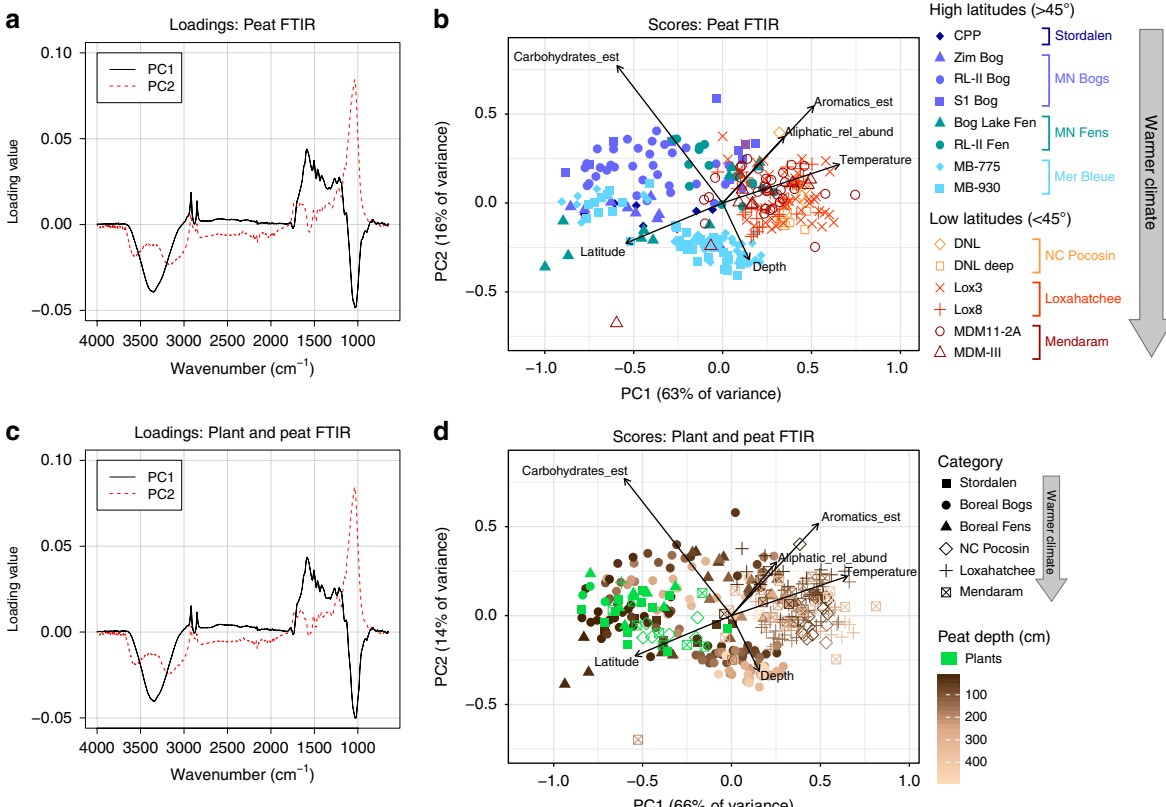

**Fig. 4** Variations in overall spectra of peat and plant samples from across the latitudinal transect. These variations are illustrated with principal components analysis (PCA) of entire Fourier transform infrared (FTIR) spectra from two sample sets: **a, b** peat samples only, with the same color scheme as Fig. 2; and **c, d** all peat and plant samples (plants shown in Fig. 5 and Supplementary Table 3), with peat color-coded by depth. Vectors on score plots indicate the direction of the increasing gradient for each variable, with arrow lengths proportional to the strength of the correlation with the PCA. All correlations were significant at $p \leq 0.001$. Due to the much larger number of peat samples ($n = 300$) compared to plant samples ($n = 39$), the clusters of points in **d** roughly correspond to those in **b**. The depth, latitude, and temperature vectors in **d** are based only on the peat samples. Carbohydrates_est; estimated % carbohydrates (as shown in Fig. 2a), aromatics_est; estimated % aromatics (as shown in Fig. 2b), aliphatic_rel_abund; aliphatic relative abundance (as shown in Supplementary Fig. 3a), temperature; mean annual temperature (°C), latitude; latitude (°N), and depth; depth below peat surface (cm)

climates. Compared to carbohydrate content, differences in aromatic content between potential source plants and peat (Fig. 5b) were less frequently significant and showed no consistent trends. Within the peat column across a broader range of depths, PCAs of the entire FTIR spectra (Fig. 4) revealed similar trends to those seen between plants and peat: the loadings of both PCAs showed that while PC1 (correlated mainly with latitude and temperature) was driven by changes in both carbohydrates and aromatics (peaks at ~1030 and ~1500 + ~1600 cm$^{-1}$, respectively), PC2 (correlated mainly with depth) was primarily driven by changes in the carbohydrate peak (~1030 cm$^{-1}$), with higher PC2 scores indicating higher carbohydrate content at shallower depths. This pattern suggests that transformation of plant material into peat and subsequent humification are driven primarily by carbohydrate loss[20].

Decomposition-induced changes in peat chemistry are driven by interactions between temperature, litter chemistry, and water saturation. Despite the more rapid decomposition that normally occurs under non-saturated, aerobic conditions[5], the non-saturated (and thus likely aerobic) CPP site at Stordalen had comparable carbohydrate and aromatic contents to the boreal Minnesota and Mer Bleue sites (Fig. 2). This lack of extensive humification at CPP may be due to the extremely cold temperatures and short growing seasons at this Arctic latitude (68°N). In contrast, the mid-latitude Mer Blue Bog (45°N), with a

water table of 30–40 cm below the surface, showed a greater decline in carbohydrates in the top ~50 cm compared to the higher water table sites in Minnesota with similar climates (Fig. 2a; Supplementary Fig. 4; Table 1). At even lower latitudes, the NC Pocosin site, also with a low water table (−30 cm at the time of sampling (Table 1), and sometimes as deep as −90 cm), had the lowest carbohydrate content in the entire data set (Fig. 2a; Fig. 3a, c) and significantly lower carbohydrate and greater aromatic content than the source plants (Fig. 5). This high degree of transformation is consistent with the unusually old age of the peat (Fig. 6), and likely reflects a combination of extensive decomposition (driven by low water tables and compounded by warm temperatures) and frequent low-intensity fires (which preferentially combust carbohydrates and produce pyrogenic aromatic compounds)[42–44]. Combined with moisture limitation of phenol oxidase activity during seasonal drought (which concentrates shrub-derived phenolics)[15], these processes create an especially recalcitrant peat that resists further mineralization, thus enabling peat accumulation despite seasonal semi-aerobic conditions down to 30–90 cm[15].

Peat radiocarbon ages (Fig. 6) suggest a wide variability in peat accretion rates at the different sites, possibly reflecting other factors (such as long-term precipitation patterns, fire, and variable hydrogeologic settings) that affect peatland development. The DNL deep core in the NC Pocosin had the oldest peat in the

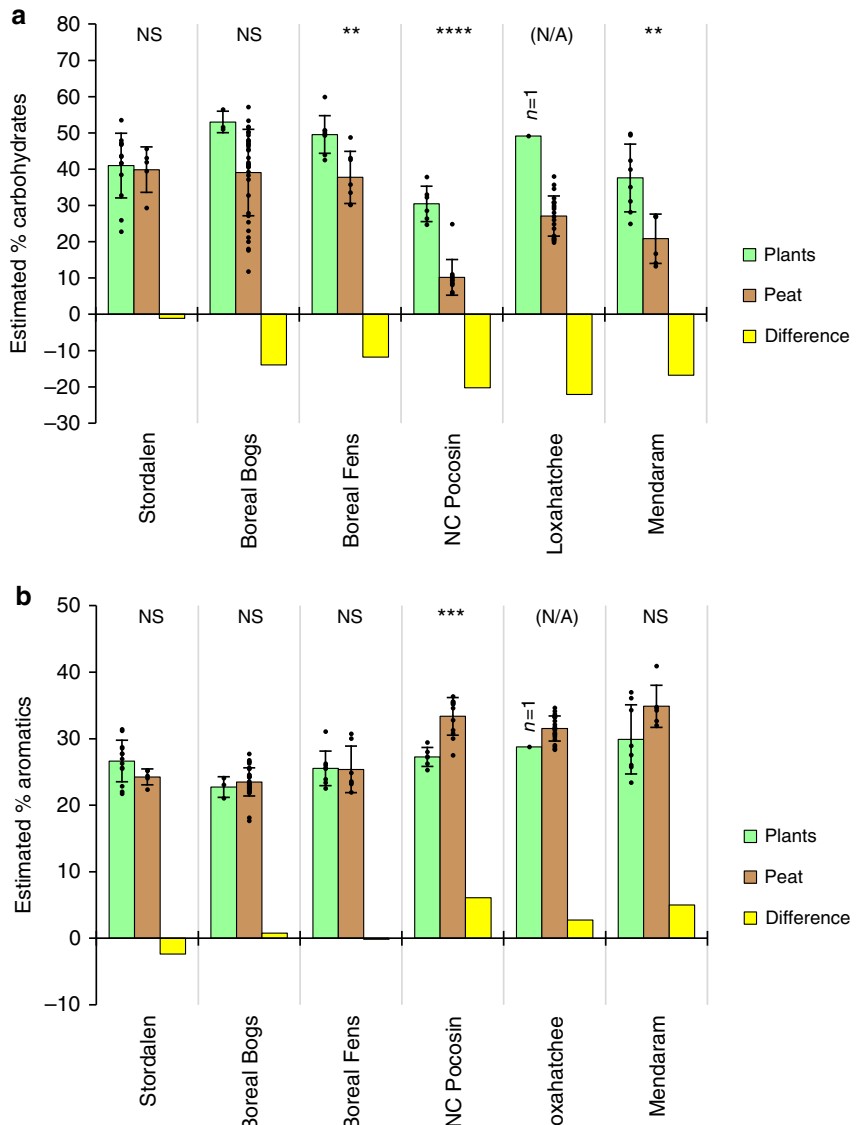

**Fig. 5** Comparisons of plant and surface peat chemistry across the latitudinal transect. Each plot shows estimated **a** carbohydrate or **b** aromatic contents in dominant plant types at each site category (green), peat from ≤50 cm at each site category (brown), and the difference (peat – plants) (yellow). For peat samples, Boreal Bogs includes MN Bogs and Mer Bleue, and Boreal Fens includes MN Fens. Error bars represent standard deviations (1 SD) of the measured samples (shown individually as points), and do not account for uncertainty in species composition of peat-forming plants. Significance of differences between plants and peat (unpaired $t$ test) are indicated with asterisks: $*p \leq 0.05$; $**p \leq 0.01$; $***p \leq 0.001$; $****p \leq 0.0001$; NS = not significant; N/A = significance could not be determined due to $n = 1$. Numbers of plant samples (Supplementary Table 3): Stordalen, $n = 13$; Boreal Bogs, $n = 3$; Boreal Fens, $n = 8$; NC Pocosin, $n = 6$; Loxahatchee, $n = 1$; and Mendaram, $n = 8$. Numbers of peat samples (Supplementary Table 2): Stordalen, $n = 5$; Boreal Bogs, $n = 35$; Boreal Fens, $n = 7$; NC Pocosin, $n = 12$; Loxahatchee, $n = 22$; and Mendaram, $n = 6$

latitudinal transect, particularly in the surface ~50 cm, where the peat was at least twice as old as peat from the same depth at other sites (Fig. 6). This extremely slow apparent peat accretion rate likely reflects periodic loss of surface peat due to fires[33], which have dramatically increased in severity due to ditching. Consistent with this greater age, the NC Pocosin peat also appeared more chemically transformed compared to the other sites, as reflected in the unusually low peat carbohydrate content and appearance as an outlier in the regressions with latitude and temperature (Fig. 3). In contrast, the tropical Mendaram site has the youngest basal peat in the latitudinal transect (Fig. 6), indicating more rapid peat accretion and carbon sequestration compared to the other peatlands[3,14]. These findings suggest that the younger peat at Mendaram is less chemically transformed

than older peat from the same depths at other sites. This interpretation is consistent with the lack of clear depth trends in the Mendaram peat cores (Fig. 2), in agreement with the slow rates of anaerobic decomposition previously observed in undisturbed tropical peatlands[3,13,14]. In addition, the rapid rate of peat accretion at this site is also a likely function of the high primary productivity[9] and continuously high rainfall[45] of the equatorial lowland tropics. Within high-latitude sites, the large RL-II Bog and Fen sites within the Glacial Lake Agassiz Peatlands (GLAP) showed more rapid peat accretion rates than the smaller S1 Bog or Mer Bleue sites. This difference reflects the importance of local hydrogeologic setting constraints on peatland formation: Peat accumulation rates in the GLAP are unusually high due to the gentle regional slopes and sparse distribution of bounding

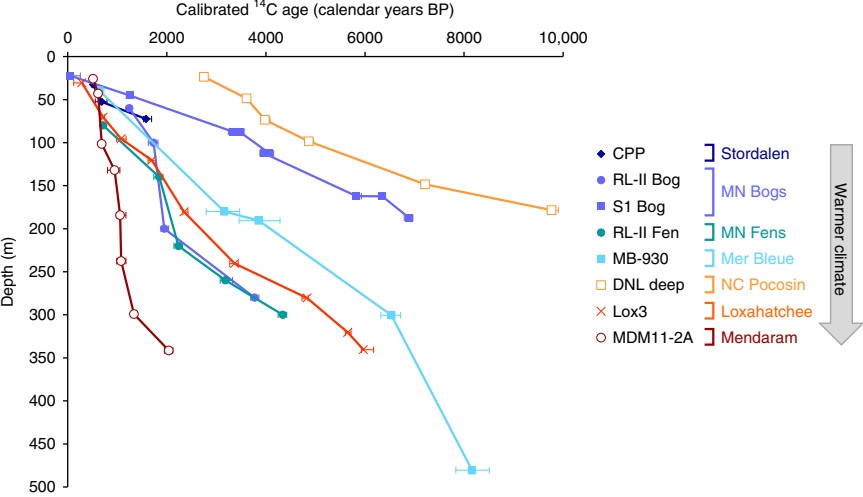

**Fig. 6** Peat radiocarbon ages in selected sites. Ages are calibrated to calendar years before present (BP). Radiocarbon ages for sites CPP, DNL deep, and Lox3 are from this study, and ages for the other sites are from the literature (references in Methods). Each point represents the median and asymmetrical 95.4% confidence interval (2σ) of the calibrated age estimates (see Methods). Error bars not visible are within the symbols

rivers, which amplify recharge and thus create ideal conditions for the rapid formation of these large peatlands[31,46]. Similarly, peat accumulation in the subtropical Loxahatchee site (northern Everglades), which began around 5000 years ago due to increased precipitation, was aided by the slowing of freshwater drainage due to the flat topography of the Everglades bedrock basin and surrounding South Florida landscape combined with long-term sea-level rise over the past 4000 years[35,47,48].

Peat ages and comparisons with plant chemistry suggest variability in the relative importance of factors leading to peat accumulation at different low-latitude sites—that is, recalcitrant plant inputs and anaerobic conditions in the Mendaram site, reduced drainage due to long-term sea-level rise in the Loxahatchee, and recalcitrance resulting from plant inputs and extensive peat transformation in the NC Pocosin site. However, despite this variability, systematic trends in peat carbohydrate and aromatic content were still observed across the entire latitudinal transect (Figs. 2, 3, and 4). This increasing recalcitrance of peat with warmer climates mirrors the increasing aromatic content in peat-forming plants towards the equator (Figs. 4d and 5b), highlighting the general importance of plant inputs for peat recalcitrance in different climatic zones. These geographic trends strongly suggest that two factors—low carbohydrate and high aromatic content—contribute to the preservation of peat in warm climates, despite differences in the relative contributions of plant inputs vs. peat transformation in driving this chemical composition.

**Implications for warming of northern peatlands**. Warming at high latitudes could stimulate peat mineralization to $CO_2$ and $CH_4$, producing a positive warming feedback[32,49–53]. This feedback may be dampened if the processes that preserve peat in southern peatlands become active at high latitudes, but the short-term and long-term climate effects depend on the exact mechanisms that are activated. If the vegetation changes towards shrubs typical of pocosins and other lower-latitude peatlands (as is expected if warming leads to drier conditions[54,55]), the increased release of plant-derived phenolics and other aromatics could prevent substantial carbon loss, possibly even inhibiting the decomposition of older *Sphagnum* and sedge peat[15]. If warmer wetter conditions favor an increased abundance of sedges (as is

predicted for systems similar to our Stordalen site[56,57]), substantial short-term decomposition and greenhouse gas release are much more likely, but this carbon loss may be balanced or exceeded by increased primary productivity[57,58]. In addition, the preferential decomposition of labile carbohydrates relative to more recalcitrant aromatics could then lead to long-term stability of catotelm peat[16,17,59]. In support of this prediction, Wilson et al.[29] found that deep peat in the S1 Bog was resistant to decomposition even after warming to 9 °C above ambient temperatures. This recalcitrance may be a result of sphagnan's inhibitory effect[27,28] and/or the already advanced humification and low carbohydrate content of deep peat at this site (Fig. 2a). As long as the peat remains water saturated and is already humified, deep peat at high latitudes may therefore be partially preserved under warming. However, warming of northern peatlands could still cause a considerable short-term positive warming feedback due to increased greenhouse gas release from decomposition of non-humified surface peat[29], plant litter[60], and dissolved organic matter[61,62]. Under scenarios of water table drawdown, the resulting aerobic conditions and higher summer temperatures at the soil surface could cause substantial short-term carbon loss as $CO_2$[52], which may then subside in the long term as high-phenolic shrubs become dominant[15,54]. Conversely, increased $CH_4$ emissions are likely if the peat continues to be water saturated[29], particularly in permafrost areas where thaw-induced subsidence leads to increased anaerobic conditions[52] and vegetation shifts toward fens that produce more labile organic matter[49–51,53].

## Methods

**Detailed site descriptions and peat sampling**. Stordalen Mire is a peat plateau in northern Sweden underlain by discontinuous permafrost, which is thawing as the Arctic warms. Stordalen includes a patchwork of habitats with varying vegetation and hydrology determined primarily by the presence of permafrost and active layer depth[50,56,63]. The site of core collection (CPP: 68.3531°N, 19.0473°E) is an aerobic, ombrotrophic permafrost palsa with an active layer depth of ~60 cm, and is vegetated by a combination of forbs, ericaceous and woody shrubs, lichen, mosses (including some *Sphagnum fuscum*), and *Eriophorum vaginatum*. These vegetation and hydrological characteristics are similar to the Palsa site described by McCalley et al.[51] and Mondav et al[64]. The specific site described here, CPP, is on the rim of a thermokarst feature which has been described previously (Hodgkins et al.[49,65]; site PHS). The CPP core was collected in June 2012 using a 6.6 cm diameter ice auger and included the active layer (0–60 cm) and permafrost peat (60–75 cm).

The S1 Bog is located in the Marcell Experimental Forest, near Grand Rapids, Minnesota. This site is characterized by hummock–hollow microtopography, with

hummocks dominated by *Sphagnum magellanicum* and hollows dominated by *Sphagnum angustifolium*, and has an overstory of black spruce (*Picea mariana*) and larch (*Larix laricina*) trees[21]. The core from this site was collected from a hollow in the T3F plot (47.5063°N, 93.4527°W), the control plot of the Spruce and Peatland Responses Under Climatic and Environmental Change (SPRUCE) experiment (http://mnspruce.ornl.gov/). This core was collected in July 2012 with a modified hole saw for surface peat (0–50 cm) and with a Russian peat corer for deeper peat (50–200 cm)[21]. Geochemical data for this core are reported in Tfaily et al[21].

Zim Bog (47.1791°N, 92.7146°W) is a strongly ombrotrophic bog dominated by *S. fuscum* with an overstory of black spruce (*P. mariana*) and ericaceous shrubs similar to the S1 Bog[66]. Peat from this site was collected in May–June 2013 with a Russian peat corer.

Bog Lake Fen (47.5051°N, 93.4890°W) is located in the Marcell Experimental Forest, ~2.7 km west of the S1 Bog, and is a weakly minerotrophic poor fen. Vegetation at this site includes a lawn of *Sphagnum* moss (mostly *Sphagnum papillosum*) mixed with dense *Eriophorum* and *Carex* sedges, with occasional shrubs (e.g., *Chamaedaphne calyculata* and *Vaccinium macrocarpon*) and northern pitcher plants (*Sarracenia purpurea*)[66]. Peat from this site was collected in May–June 2013 using a Russian peat corer.

Red Lake II Bog (RL-II Bog; 48.2547°N, 94.6976°W) and Red Lake II Fen (RL-II Fen; 48.2897°N, 94.7083°W) are located in the GLAP[31,67]. RL-II Bog is located on the forested crest of a large bog complex. The peat surface is carpeted with continuous *Sphagnum* spp. and is forested with black spruce and ericaceous shrubs. Water drains from the bog across an unforested *Sphagnum* lawn, collecting in narrow fen water tracks. The upper 3 m of the RL-II Bog core is comprised of woody *Sphagnum* peat, which is underlain by 1.3 m of woody fen peat with decomposed sedge biomass and Amblystegiaceae moss remains. RL-II Fen has standing water and is dominated by sedges including *Carex lasiocarpa*, *Carex limosa*, *Rhynchospora alba*, and *Rhynchospora fusca*[61]. The upper ~80 cm of peat are comprised of sedge and *Scorpidium* moss remains, with bog deposits containing *Sphagnum* and woody peat below a depth of 100 cm. Cores from both RL-II Bog and RL-II Fen were collected in 2009 using a modified Livingstone piston corer with a 4-in. steel barrel and a serrated cutting edge[68].

Mer Bleue Bog (45.4°N, 75.5°W) is a cool temperate ombrotrophic bog located ~10 km east of Ottawa, Ontario, Canada. The bog is fairly dry, with a summer water table 30–40 cm below the surface, and has a hummock–hollow microtopography. The ground is carpeted by *Sphagnum* spp. (*S. capillifolium* and *S. magellanicum*), with an overstory of shrubs (*Chamaedaphne calyculata*, *Ledum groenlandicum*, *Kalmia angustifolia*, and *Vaccinium myrtilloides*), sparse sedges (*E. vaginatum*), and a few small trees (*P. mariana*, *Larix laricina*, and *Betula populifolia*) on hummocks[69]. Two cores were collected with a Russian peat corer in summer 1998: MB-775 between the bog center and edge, and MB-930 (described in ref. [69]) near the center of the bog.

The Pocosin Lakes National Wildlife Refuge (35.7°N, 76.5°W) is a shrub-dominated peatland located in eastern North Carolina, USA. Since this peatland is not inundated, peat decomposition is thought to be inhibited by phenolics released by shrubs[15] and refractory black carbon produced by the frequent low-intensity fires at this site[33,70,71]. At the coring locations, the most recent fire occurred 30 years prior to sampling in May 2015. The DNL core was collected with a home-built 10-cm-wide stainless steel square peat corer with a removable cutting panel inserted last to cut the fourth side, which reduces peat compaction. The DNL deep core was collected with a Russian peat corer.

The Loxahatchee National Wildlife Refuge is an inundated mesotrophic peat marsh in the northern Everglades with a peat depth of ~3 m. Both of the sites sampled for this study (Lox3: 26.597°N, 80.357°W and Lox8: 26.520°N, 80.335°W) have 0.5–1 m of standing water above the peat surface and are vegetated primarily by *Cladium jamaicense*, with tree islands within 10 m of the coring locations. Cores from both sites were collected with a Russian peat corer in October 2015.

The Mendaram study site is a pristine tropical peat swamp forest located in the Ulu Mendaram Conservation Area in the Belait District of Brunei Darussalam, northwest Borneo. This site is dominated by *Shorea albida* trees with a dense understory of *Pandanus andersonii*. Peat is comprised of a combination of woody debris, which resists decomposition due to its coarse physical structure and lignin content[11–14,72], and the remains of leaves and non-woody plants that accumulate in flooded tip-up pools left by fallen trees[10,73]. Due to a combination of year-round biomass production, wet climate with consequent anaerobic conditions, and the recalcitrant nature of the peat, undisturbed Southeast Asian coastal peat domes such as the Mendaram have the world's greatest long-term carbon accumulation rates per unit area[3,14], but this carbon sink is becoming a source as these peatlands are drained and burned for agriculture[3,13,14]. For this study, two cores were collected from the Mendaram site. The first core, MDM11-2A (4.3727°N, 114.3550°E), was collected on 1 November 2011 using a Livingstone piston sampler with a 10-cm-wide core barrel and a serrated cutting edge. The sampling procedure and physical and chemical properties for this core are described in detail by Dommain et al[10]. A second core, MDM-III (4.3702°N, 114.3542°E; within the undisturbed site described by Cobb et al. [45] and close to Mendaram Site III sampled by Gandois et al.[11]), was collected in 2014 with a 5-cm-diameter Eijkelkamp Russian peat corer. Due to the cores' heterogeneity, we omitted data from the core sections composed mainly of fresh wood fragments (20–21 and 110–111 cm in the MDM11-2A core), which were poorly decomposed and thus obscured the signal of peat humification with depth[10].

**Plant sampling.** The plants collected for comparison with peat, including species, plant parts, site of collection, and other metadata, are listed in Supplementary Table 3.

For the NC Pocosin and Loxahatchee plants, each sample represents a composite of several samples from the same species, which were mixed into one combined sample. Thus, although $n = 1$ for Loxahatchee plants (*Cladium jamaicense*), this sample can be considered representative of this species because it is a composite of several individual plants.

For the *Shorea albida* leaves, intact leaf litter was collected in lieu of fresh leaves due to the difficulty of obtaining leaves directly from the tall trees. *Shorea albida* wood blocks were cut from discarded lumber at the former Lutong Sawmill, 8 km from the coring sites. These blocks were collected as part of a separate wood decomposition experiment, in which blocks (approx. $2.2 \times 4.5 \times 5.5$ cm$^3$) were buried in the peat in February 2012 and harvested in August 2015. % Mass loss was obtained based on masses before and after burial. Since no unburied blocks were saved for FTIR analysis, the buried blocks with the lowest % mass loss were used for comparison with peat. These blocks had been buried at depths of 143, 141, 84, and 83 cm, and had relatively small mass losses of 2.22, 2.19, 2.51, and 1.54%, respectively.

**Radiocarbon ages.** Peat samples from the CPP, DNL deep, and Lox3 cores were prepared for radiocarbon analysis using the methods of Corbett et al[67]. Dried and ground peat, cupric oxide, copper shots, and silver were added to combusted quartz tubes. The tubes were evacuated and flame-sealed on a vacuum line. The organic matter was then converted to $CO_2$ gas by combusting the tubes for 18 h at 850 °C [53]. The $CO_2$ was then cryogenically purified and sealed into another Pyrex tube on the vacuum line. The tubes of purified $CO_2$ were then sent to the National Ocean Sciences Accelerator Mass Spectrometry Facility for analysis of $^{14}$C. Radiocarbon ages were then calculated based on a $^{14}$C half-life of 5568 years. For the other sites, radiocarbon ages were obtained from the following literature sources: RL-II Bog and RL-II Fen, refs. [61,75]; S1 Bog, ref. [21]; Mer Bleue, ref. [69]; and Mendaram, ref [10].

Radiocarbon ages were calibrated to calendar years with OxCal Online (University of Oxford, https://c14.arch.ox.ac.uk/oxcal/OxCal.html), using the IntCal13 calibration curve. Ages in Fig. 6 are reported as median ages before present (BP), with error bars as asymmetrical 95.4% confidence intervals. Depths with age reversals were ignored in the age-depth models.

**Fourier transform infrared spectroscopy.** Our estimations of carbohydrate and aromatic contents are based on a newly developed analysis technique for FTIR spectra. FTIR is a common spectroscopic method for analyzing the composition of solid-phase organic matter. When used with attenuated total reflectance (ATR), this method is relatively fast and inexpensive, but is not fully quantitative. FTIR data are typically analyzed either qualitatively by changes in the shape of the spectra, or semi-quantitatively with ratios of peak heights, most commonly humification indices (i.e., ratios of aromatic:carbohydrate or aliphatic:carbohydrate peaks)[19,21,49,76–79]. A disadvantage of humification indices, as with other peak ratios, is that it is difficult to discern whether humification is driven by changes in carbohydrates vs. aromatics and aliphatics.

Other studies have overcome this problem by correlating FTIR data with wet chemistry-based measurements of carbohydrates, lignin, lipids, proteins, and other compounds, allowing FTIR to be used more quantitatively. These studies include simple calibrations with ratios of peak heights[76,77,79], as well as more complex multivariate techniques such as partial least squares[80–82]. However, neither of these techniques provide a basis for estimating relative abundances of individual compounds, apart from ratios, that are not directly calibrated. Some studies have isolated individual peaks and correlated them to wet chemistry[83], but this technique is relatively uncommon.

In this study, we introduce a new FTIR data processing method that allows for more thorough quantification of compound classes. First, instead of normalizing peak heights relative to other peaks via humification indices, we better isolate each compound by instead normalizing peaks to the integrated area of the entire spectrum. Next, we use a set of calibration standards to compare these normalized peak heights with wet chemistry analyses, specifically % cellulose + hemicellulose (carbohydrate peak, ~1030 cm$^{-1}$) and % Klason lignin (aromatic peaks, ~1510 and ~1630 cm$^{-1}$)[22], and show that these measures are linearly correlated. Thus, this study not only provides a method for estimating concentrations of carbohydrates and aromatics, it also suggests that other area-normalized peak heights may be interpretable as relative abundances for cross-sample comparison of individual compound classes.

In preparation for FTIR analysis, peat and plant samples were freeze-dried and then ground to a fine powder for 2 min using a SPEX SamplePrep 5100 Mixer/Mill ball grinder. Calibration standards (for description, see Calibration of FTIR data) were dried at 50 °C, ground in a Wiley mill to pass through a 60-mesh screen, and re-dried to constant weight at 50 °C[22]. FTIR spectra were collected with a PerkinElmer Spectrum 100 FTIR spectrometer fitted with a CsI beam splitter and a deuterated triglycine sulfate detector. Transmission-like spectra were obtained with a Universal ATR accessory with a single-reflectance system and made from a zinc selenide/diamond composite. Samples were placed directly on the ATR crystal, and force was applied so that the sample came into good contact with the crystal.

Spectra were acquired in % transmittance mode between 4000 and 650 cm$^{-1}$ (wavenumber) at a resolution of 4 cm$^{-1}$, and four scans were averaged for each spectrum. The spectra were ATR-corrected to account for differences in depth of beam penetration at different wavelengths, and then baseline-corrected, with the instrument software. Spectra were then converted to absorbance mode for subsequent data analysis.

**FTIR peak finding**. Since our study was focused on carbohydrate and aromatic contents of peats, we chose to isolate specific FTIR bands representative of those two functional groups and quantitate based on the peak heights of those bands. We recognize that multivariate analyses of entire FTIR spectra (such as partial least squares) can identify additional spectral features related to humification[80], but here we are quantifying carbohydrate and aromatic content using FTIR spectra correlated with wet chemical methods that only provide data on these two functional groups.

Due to differences in sample chemistry, the exact locations of target FTIR peaks varied between samples, so the locations of peaks and peak endpoints (Supplementary Fig. 1) were individually determined for each sample. Peak endpoints were first found based on local minima in the expected region of each peak endpoint (Supplementary Table 1), or based on the maximum of the second derivative if there was no local minimum. Absorbances between the peak endpoints were then baseline-corrected by subtracting the absorbance below a baseline drawn between the endpoints of each peak[78,79] (Supplementary Fig. 1). Exact peak locations were then found based on the maximum baseline-corrected absorbance between the peak endpoints. Finally, to account for matrix-induced and instrument-induced variations in overall absorbance between samples, the baseline-corrected peak heights were divided by the total integrated area of the spectrum to give normalized corrected peak heights. These calculations were performed with a custom script in R (version 3.3.2).

To avoid interference from silicates, which produce a large FTIR peak that interferes with the carbohydrate peak (~1030 cm$^{-1}$)[84], core depths that contained silicates (determined based on the presence of peaks at 3695 and 3620 cm$^{-1}$ (kaolinite) and/or 780 cm$^{-1}$ (silicate minerals), combined with a large peak at ~1030 cm$^{-1}$) were excluded from our analysis.

**Calibration of FTIR data**. The calibration standard set was comprised of 58 plant and paper samples, which included hardwoods, softwoods, leaves and grasses, needles, old corrugated cardboard, old newsprint, old magazines (OMG), and office paper (OFF)[22]. Weight percentages of cellulose + hemicellulose (determined by acid hydrolysis and high-performance liquid chromatography after rinsing with toluene and ethanol) and Klason lignin (acid-insoluble material minus ash), previously measured by De La Cruz et al.[22], were used to calibrate FTIR absorbances arising from carbohydrates (carb, ~1030 cm$^{-1}$) and aromatics (arom15, ~1510 cm$^{-1}$; and arom16, ~1630 cm$^{-1}$), respectively. It is important to note that since Klason lignin is operationally defined as the fraction of material that is acid insoluble minus ash, it includes not just structural lignin, but also other aromatics such as tannins, other non-lignin-derived polyphenols, and biochar.

In the calibration data set, the normalized corrected peak heights (*carb* for carbohydrates, and arom15 and arom16 for aromatics; Supplementary Fig. 1) were compared to the wet chemistry methods (% cellulose + hemicellulose for carbohydrates and % Klason lignin for aromatics) by linear regression ($n = 54$ for both regressions; Supplementary Fig. 2). The carbohydrate calibration omitted the OMG samples (4 samples out of 58) because these had a large peak that overlapped with the carb peak, likely arising from clay coatings used to produce a glossy finish on the magazine paper[84,85] (Supplementary Fig. 8). The aromatic calibration omitted the OFF samples (4 samples out of 58) because they represent a chemical pulp that has most of its lignin removed during processing[85]. Despite these unusual spectral features in the OMG and OFF standard sets, they still fit with the other standards (with spectra more similar to our peat samples; Supplementary Fig. 8) along the aromatic and carbohydrate calibration curves, respectively (Supplementary Fig. 2).

The FTIR peak heights in the calibration sample set were in good agreement with measured % cellulose + hemicellulose and % Klason lignin. For carbohydrates, % cellulose + hemicellulose was significantly correlated with the carb FTIR peak ($R^2 = 0.80$, $p < 0.0001$; Supplementary Fig. 2a). For aromatics, three regressions were performed using either the arom15 peak, the arom16 peak, or the sum of both peaks as the *x*-variable, so that the regression with the best fit could be selected for subsequent analysis. Among these, the sum of both peaks (arom15 + arom16) produced the best correlation ($R^2 = 0.58$, $p < 0.0001$; Supplementary Fig. 2b) and was therefore used for further analysis of aromatic content. These fits, which used area-normalized and baseline-corrected peak heights, were also slightly better than the same ones performed on area-normalized peaks without the baseline corrections (carbohydrates: $R^2 = 0.80$; aromatics: $R^2 = 0.38$).

Carbohydrate and aromatic contents in peat and plants were estimated based on their FTIR carb, arom15, and arom16 peak heights, using the regression equations shown in Supplementary Fig. 2 as calibration curves. Standard errors of the *y* estimate for each calibration (two-tailed: SE = 9 for carbohydrates, and SE = 5 for aromatics) were used as the standard errors for estimated % carbohydrates and % aromatics in each sample (Fig. 2; Supplementary Fig. 4).

The strength and linearity of both calibrations (Supplementary Fig. 2) demonstrates that carbohydrate content can be estimated with the carb FTIR peak

(~1030 cm$^{-1}$), while aromatic content can be estimated with the sum of the arom15 and arom16 FTIR peaks (~1510 and ~1615 cm$^{-1}$). This is despite the considerable variation in spectral features of the standards used in this study (Supplementary Fig. 8), which would have complicated the interpretation of whole-spectra regression techniques such as partial least squares. However, our approach should only be used if there are no compounds that produce large peaks that overlap with the specific peaks being analyzed (e.g., silicates that overlap with the carbohydrate peak), which were not in this study (OMG was omitted from the carbohydrate calibration and OFF was omitted from the aromatic calibration).

More broadly, our study suggests that even for uncalibrated compound classes (such as aliphatics (Supplementary Fig. 3) and organic acids), normalization of peak heights to spectral area (as was done in this study) may provide a means of estimating a compound's relative abundances across samples without normalization to any other single FTIR peak (as is the case with the commonly used humification indices[18,19,21,49,76–79]). When calibrated with wet chemistry, these relative abundances can become fully quantitative, allowing the measurement of compound concentrations in a large number of samples without the need for more labor-intensive wet chemistry methods.

**Statistical analysis**. Calibration of the FTIR data, and subsequent estimation of % carbohydrates and % aromatics in plants and peat, are described in the preceding section (Calibration of FTIR data).

Overall depth trends for carbohydrate, aromatic, and aliphatic content in high-latitude and low-latitude peatlands (Fig. 2c, d; Supplementary Fig. 4b) were visualized with locally weighted polynomial regression (LOESS). The curves were plotted with the built-in geom_smooth() function of the ggplot2 R package (version 1.0.0; built using R version 3.0.3), grouping the points by high and low latitudes (separated by the midpoint between pole and equator, 45°N), and using the default LOESS() function settings: polynomial degree = 2, $\alpha$ = 0.75 ($\alpha$, or span, is the fraction of points used to fit each local regression), and shaded errors = 95% confidence interval of the smooth line.

Trends in carbohydrate and aromatic content with latitude and mean annual temperature (Fig. 3) were assessed using linear regressions. In these regressions, each point represents an average ± 1 SD of the samples from depths ≤50 cm in each core (Supplementary Table 2).

For comparison of plant and peat chemistry, both sample sets were divided into categories of Stordalen, Boreal Bogs, Boreal Fens, NC Pocosin, Loxahatchee, or Mendaram (Fig. 5), based on the site classification for peat samples and typical peatland environments for plant samples (Supplementary Table 3). Each set of error bars represents 1 SD, which describes the observed variability independent of sample size, and does not account for uncertainty in the vegetation composition of peat-forming plants. Significance of differences between plants and peat in the same category was assessed with unpaired two-tailed *t* tests.

For PCA, FTIR spectra were preprocessed by scaling the absorbances such that the integrated area of each spectrum was a constant value of 100. PCA was then performed in R (version 3.4.4) with the prcomp function. External variables were fitted to the PCAs using the envfit function in the vegan package (version 2.5–1; ref. [86]) and plotted as vectors in the score plots (Fig. 4b, d). When fitting depth, latitude, and temperature to the PCA of plants and peat (Fig. 4d), plant samples were excluded from the vector fits because they do not have depths, and their origin latitudes do not correspond exactly with the peat samples against which they are compared (Supplementary Table 3).

**Code availability**. The R script used for the analysis of FTIR spectra, including a tutorial, is available at https://github.com/shodgkins/FTIRbaselines (permanent link to the version used in this study: https://github.com/shodgkins/FTIRbaselines/tree/175a18c5ecafb472d5b6a3648506dd171ecca37c).

## Data availability

The FTIR spectra and R script output data, including exact locations of peaks, baseline endpoints, and peak heights, are available as Supplementary Excel files (calibration standards: Supplementary Data 1; peat: Supplementary Data 2; plants: Supplementary Data 3). The measured % cellulose + hemicellulose and % Klason lignin in the calibration data set are taken from De la Cruz et al.[22], and are included in Supplementary Data 1. Radiocarbon ages, both uncalibrated and calibrated, are available in Supplementary Data 4.

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

## Acknowledgements

This work was primarily funded by the US Department of Energy Office of Biological and Environmental Research under the Terrestrial Ecosystem Sciences program, under award DE-SC0012272 to Duke University and Florida State University. S.B.H. received additional funding from the NASA Interdisciplinary Studies in Earth Science program (Award # NNX17AK10G) and the US Department of Energy Office of Biological and Environmental Research under the Genomic Science program (Award DE-SC0016440). R.D. acknowledges financial support by Geo.X, the Research Network for Geosciences in Berlin and Potsdam. Funding for the sample collection and analysis at Stordalen was provided by the US Department of Energy Office of Biological and Environmental Research under the Genomic Science program (Awards DE-SC0004632 and DE-SC0010580). The S1 Bog was sampled and analyzed as part of the Spruce and Peatland Responses Under Climatic and Environmental Change (SPRUCE) experiment funded by the US Department of Energy, Office of Science, Office of Biological and Environmental Research, contract # DE-SC0012088. Collection of the Red Lake II cores was funded by NSF Award 0628647. Mer Bleue core collection was funded by the Natural Sciences and Engineering Research Council of Canada. Sample collection and radiocarbon dating of the Ulu Mendaram Conservation Area cores was supported by the National Research Foundation Singapore through the Singapore-MIT Alliance for Research and Technology's Center for Environmental Sensing and Modeling interdisciplinary research program, and by the USA National Science Foundation under Grant Nos. 1114155 and 1114161 to C.F.H. We thank the Brunei Darussalam Heart of Borneo Center and the Brunei Darussalam Forestry Department for facilitation of field work and release of staff. Rahayu Sukmaria binti Haji Sukri, Watu bin Awok, Azlan Pandai, Rosaidi Mureh, Muhammad Wafiuddin Zainal Ariffin, Laure Gandois, Jangarun Eri, Fuu Ming Kai, Kamariah Abu Salim, Nur Salihah Haji Su'ut, Amy Chua, Jeffery Muli Incham, Haji Bohari bin Idi, and Ramasamy Zulkiflee assisted with collection of the Ulu Mendaram cores. We acknowledge the LacCore facility for their support and permanent core storage. We also thank the Abisko Scientific Research Station near Stordalen Mire, the USDA Forest Service at Marcell Experimental Forest, staff at the Pocosin Lakes National Wildlife Refuge, and the USFWS at Arthur R. Marshall Loxahatchee National Wildlife Refuge for access to these field sites and logistic support. Eun-Hae Kim, Tyler Logan, Carmody McCalley, and Kristina Solheim assisted with collection of the CPP core. Malak Tfaily provided FTIR data for the Zim Bog, S1 Bog, and Bog Lake Fen cores. Joseph Portio and Penn Carnice assisted with sample preparation and FTIR analysis, and Samantha Bosman assisted with radiocarbon analysis. Rachel Wilson provided helpful suggestions on the text. We thank Annika Kristoffersson for providing the temperature data from Abisko, Sweden. Earth surface temperature data in Fig. 1 were obtained from the NASA Langley Research Center Atmospheric Science Data Center Surface meteorological and Solar Energy (SSE) web portal supported by the NASA LaRC POWER Project.

## Author contributions

C.J.R., W.T.C. and J.P.C. obtained funding; J.P.C., C.J.R., H.W., S.B.H., R.D. and W.T.C. designed research; S.B.H, C.J.R, R.D., P.H.G., N.F., H.W., M.H., A.M.H., C.F.H., A.R.C., V.I.R., S.R.V., M.A.H., P.J.H.R., and J.P.C. assisted with field work; M.M., S.R.V., and T.R.M. assisted with sample preparation; F.B.D. provided calibration standards; S.B.H., B.V., B.R.W. and M.M. performed FTIR analysis; J.T. and R.H. helped supervise FTIR analysis; S.B.H., R.D., B.V. and B.R.W. analyzed data; S.B.H. wrote the text with input from C.J.R., R.D., P.H.G., N.F., H.W., M.H., A.M.H., C.F.H., A.R.C., M.A.H., T.R.M., F.B.D., W.T.C. and J.P.C.; J.P.C., C.J.R. and W.T.C. jointly supervised the project.
