## [Peer Review File · Nature Communications]

Reviewers' comments:

Reviewer #1 (Remarks to the Author):

Here, the investigators demonstrate correlative patterns among intrinsic peat properties (indexes of carbohydrates and "aromatics"), temperature, and latitude, and ascribe changes in peat accumulation rates to these patterns. In effect, the observational study provides data in support of a compelling hypothesis as to why there is peat in warm climates. The study combines detailed characterization of peat and source plant materials from a wide range of climates with extensive radiocarbon analysis of peat C accumulation. The study is well written, and the methods clearly described. I enjoyed reading this manuscript! I think this manuscript is appropriate for the readership of Nature Communications. I offer minor comments in revision.

1) The words "Estimated carbohydrates" and "aromatics" are used somewhat generically in this manuscript. Would it be possible to provide some minor stoichiometric description of what these components are? An example of this might be particular H:C or O:C cutoffs for the compound classes (e.g., Kellerman et al. 2015, Nat. Geosci. DOI: 10.1038/NGEO2440).

2) The authors discuss a latitude-driven gradient in temperatures, and selective loss of carbohydrates. A major preservation mechanism for carbon in peatlands of course is the persistence of a high water table. The authors do mention "moisture limitation" [line 197], but it seems a major omission to not discuss water residence time or even water table fluctuations across these widely varying peatland types. Is not water table position, or water residence time, important for carbon accumulation in bogs, fens, or pocosins? How does biomolecular character change with water table fluctuations? Fires are discussed, but the mineralization capacity of aerobic heterotrophs, when the water table is low, seems to me way more important.

Reviewer #2 (Remarks to the Author):

This paper covers a very interesting area, and involves a considerable amount of field work and analysis, but I have some fundamental issues with it.

I do not find the paper easily readable and feel that it is overly complicated in style, but with important factors missing from the discussion. In addition, I have concerns about some of the methods used and their influence on the resulting conclusions. There are a lot of aspects which I wish to discuss, but may not be able to cover everything.

The key points relate to:

- Absence of values for or discussion of aliphatic or "waxy" components, which are a key feature of peat and tend to increase down the profile, and will play a significant part in how decomposable the peat is (Leifeld et al. 2012)
- No explanation as to why the direct regression of the FTIR spectra against wet chemistry analysis was not used, or reasons for using the more complex method developed here using R
- No explanation as to how the estimated errors for the composition of plants at each site can be presented with errors often as small as those for the peats (Fig. 3) given the arbitrary nature of deciding which plants and in which proportions may have contributed to peat formation (and the additional uncertainty about changes in climate and plant populations over time).
- Not using the FTIR dataset to its full potential, and missing out on important information which could be obtained by further interpretation and comparison of the spectra

FTIR analysis

The FTIR spectra could have been directly regressed against the wet chemistry results to predict the carbohydrate and aromatic content. This would have been a simpler approach and does not rely on pre-selecting which peaks to use. It seems to me that the R script for location of the peaks is unnecessary and may lead to poorer results than the direct regression (which allows the peaks which describe the observed variance to be selected). At the very least, a direct regression should have been done as a comparison to their approach to demonstrate that there is something to be

gained from it.

The calibration dataset ought to be appropriate for the population to be analysed to ensure a robust calibration and in this case it does not appear so to me, particularly for the peat analysis. I would like to see evidence that the spectra of the unknown samples being analysed fall within the spectral range of the calibration set. I also find it slightly bizarre that you would choose to use magazines for this.

A peak at 780 cm⁻¹ could arise from amorphous silica but is more likely to arise from oxalate, particularly in plant material. There is a doublet at 779 and 797 cm⁻¹ which is due to the presence of quartz, but there would not be a peak at this frequency for clay minerals, which are very likely to occur. These should have been taken into account by looking for peaks in the OH stretching region (e.g. 3695 and 3620 cm⁻¹ for kaolinite).

The FTIR spectra give the overall chemistry of the peat samples and I think a lot more could have been made of interpretation of the spectra. Where differences or similarities are described I would expect to see some illustrations. I am not comfortable with papers, using FTIR analysis, where spectra aren't shown. A lot of relationships could have been explored by looking at the spectra in more detail: For example a comparison between the peat samples from different latitudes with similar carbohydrate and aromatic analysis results could have shown whether there were other chemical differences apparent. Also, a look at the progressive changes in the peat spectra as you went down a profile could have illustrated whether the trends reported in Fig. 1 are clearly visible in the spectra. A comparison between the vegetation and the peat spectra would also be informative.

I am also concerned that only four scans were averaged per sample as this is far fewer than would be recommended and risks producing spectra with poor signal to noise ratios.

Peat chemistry and relationship to degradation

A large part of the change observed down a peat profile, in my experience, is the development of strong waxy type profile from long chain aliphatics – this is absent in this paper beyond showing aliphatic peaks in the R script. This appears to be a major omission, with no apparent justification. Papers by Leifeld indicate that the O-C % is important, rather than just the ratio between carbohydrates and aromatics (Leifeld et al. *Geophysical Research Letters*, Vol 39, L14704, 2012; Bader et al. *Biogeosciences Discuss* 2017)

There does not appear to be much attention paid to possibilities of changes in vegetation and rates of degradation with time and climatic conditions – beyond considering the water table. It is possible to come across pockets with high % carbohydrate buried deep in the peat, given the right circumstances. Analysis of macrofossils can provide information about changes in the vegetation which contribute to peat formation over time. (D. Mauquoy and D. Yeloff, *Biodivers Conserv* (2008) 17:2139–2151)

Age of the peat – for Pocosin it appears that there is no new peat and so comparisons with the top of other peatlands probably aren't relevant and may explain results for this site having values lying off the lines in the diagrams in Fig. 2. I question whether this is a valid comparison as in Leifeld et al. 2012 they suggest that a peat of ~4000 years age may have originally been buried at a depth of 2m, and this would make sense because it wasn't created recently.

Greater changes would be expected at higher temps in the low latitudes initially which would lead to the degraded more recalcitrant peat – I would like to see how these spectra compared to lower depth high latitude peat.

In Fig. 3 the composition of plants at each site is presented with errors often as small as those for the peats which I find difficult to accept, given the arbitrary nature of deciding which plants and in which proportions may have contributed to peat formation. This is without the additional uncertainty introduced by changes in climate and plant populations over time. We already know that at the Pocosin site, the peat was created thousands of years ago. I remain to be convinced that the values for the vegetation of the % carbohydrate and % aromatic content can be quoted this accurately and I would like to see how the values were arrived in order to determine how valid a comparison is it with the peat.

The number of sites and latitudes represented is not large and there are not multiple sites at given latitude and so I would like to have evidence for the representativeness of the sites and the

justification for making to make generalisations about latitude.

There appear to be some contradictions in the text in that it is stated that the southern latitudes have woody inputs and therefore they have higher aromatics but then for Loxahatchee say it is because of lignin in the roots? I would like to see a comparison of the spectra (as mentioned earlier) of the peats with the same proportions of aromatics and carbohydrates from the upper levels of southern latitudes and lower levels of northern latitudes to show that they are different and haven't just degraded quicker with the higher temperature. Correlations with temperature are shown to be slightly better than correlation with latitude (Fig. 2) so could it be the key driver?

Reviewer #3 (Remarks to the Author):

<- anonymous review ->

This is a very interesting and novel manuscript that compares the composition of boreal peat bogs with those at low latitudes (as well as surface peat vs deeper peat). The results have potential implications for peat loss in high-latitude peatlands and thus for modelling future carbon emissions. Therefore they could be of relevance for investigations of future climate change.

You are mainly using space (latitude) to make your points. Given the large amount of palaeo work done on peat bogs, could the time component be discussed more? E.g., see Charman et al., 2013. Climate-related changes in peatland carbon accumulation during the last millennium. *Biogeosciences* 10, 929-944

My main problem with the manuscript is its sample size - only five regions (most with multiple sites) are reported. Can the results from this small dataset really be extrapolated and generalised? Can the vegetation chemistry really be concluded to be that constant over similar latitudes? Such questions should at least be discussed in the text. This study should perhaps be seen as a valid first step to a wider analysis of the found patterns.

How thick are the studied deposits? Were all cores sampled until reaching the bog's bottom? If not, would you expect any found trends to have continued further down?

Fig. 1: this spaghetti-diagram is very hard to read. Consider separating into more panels, e.g. a panel per site showing both %carbohydrates and %aromatics.

We need a map. This map could show the studied regions as well as where peatlands of different types can be found.

Line 151, what family does C. stand for here?

Line 426, do you mean that no age-reversals occurred, or that some ages were ignored to avoid age reversals?

Rows 11-19 in the 14C xlsx file - Why do these values have so many decimals? Is this because you estimated ages from a plot?

Sufficient detail is provided in the methods to enable replication of this study.

Response to Reviews

(Reviewer comments in normal text; author response in **teal bold**)

Reviewer #1 (Remarks to the Author):

Here, the investigators demonstrate correlative patterns among intrinsic peat properties (indexes of carbohydrates and “aromatics”), temperature, and latitude, and ascribe changes in peat accumulation rates to these patterns. In effect, the observational study provides data in support of a compelling hypothesis as to why there is peat in warm climates. The study combines detailed characterization of peat and source plant materials from a wide range of climates with extensive radiocarbon analysis of peat C accumulation. The study is well written, and the methods clearly described. I enjoyed reading this manuscript! I think this manuscript is appropriate for the readership of Nature Communications. I offer minor comments in revision.

Response: We thank the reviewer for their constructive comments.

1) The words “Estimated carbohydrates” and “aromatics” are used somewhat generically in this manuscript. Would it be possible to provide some minor stoichiometric description of what these components are? An example of this might be particular H:C or O:C cutoffs for the compound classes (e.g., Kellerman et al. 2015, Nat. Geosci. DOI: 10.1038/NGEO2440).

Response: We have added text (li. 110-113) defining what these fractions are. These definitions are based on the wet chemistry methods used to calibrate the FTIR peaks, and are copied below:

‘Based on the techniques used to calibrate the FTIR peaks (see Methods; ref. 39), “carbohydrates” includes acid-hydrolysable polysaccharides, whereas “aromatics” includes lignin and other unsaturated acid-insoluble material such as tannins and humic substances.’

2) The authors discuss a latitude-driven gradient in temperatures, and selective loss of carbohydrates. A major preservation mechanism for carbon in peatlands of course is the persistence of a high water table. The authors do mention “moisture limitation” [line 197], but it seems a major omission to not discuss water residence time or even water table fluctuations across these widely varying peatland types. Is not water table position, or water residence time, important for carbon accumulation in bogs, fens, or pocosins? How does biomolecular character change with water table fluctuations? Fires are discussed, but the mineralization capacity of aerobic heterotrophs, when the water table is low, seems to me way more important.

Response: We do discuss differences in water content between the sites. We realize these discussions could have gotten lost in the discussion of peat chemistry, and accordingly, we have clarified the text to be more explicit about the effects of water content on the observed trends. The revised text (li. 220-239) is copied below:

‘Decomposition-induced changes in peat chemistry are driven by interactions between temperature, litter chemistry, and water saturation. Despite the more rapid decomposition

that normally occurs under non-saturated, aerobic conditions⁵, the non-saturated (and thus likely aerobic) CPP site at Stordalen had comparable carbohydrate and aromatic contents to the boreal Minnesota and Mer Bleue sites (Fig. 1). This lack of extensive humification at CPP may be due to the extremely cold temperatures and short growing seasons at this Arctic latitude (68 °N). In contrast, the mid-latitude Mer Blue Bog (45 °N), with a water table of 30–40 cm below the surface, showed a greater decline in carbohydrates in the top ~50 cm compared to the higher-water-table sites in Minnesota with similar climates (Fig. 1a; Supplementary Fig. 4; Table 1). At even lower latitudes, the NC Pocosin site, also with a low water table (–30 cm at the time of sampling [Table 1], and sometimes as deep as –90 cm), had the lowest carbohydrate content in the entire dataset (Fig. 1a; Fig. 2ac) and significantly lower carbohydrate and greater aromatic content than the source plants (Fig. 3). This high degree of transformation is consistent with the unusually old age of the peat (Fig. 4), and likely reflects a combination of extensive decomposition (driven by low water tables and compounded by warm temperatures) and frequent low-intensity fires (which preferentially combust carbohydrates and produce pyrogenic aromatic compounds)^{57–59}. Combined with moisture limitation of phenol oxidase activity during seasonal drought (which concentrates shrub-derived phenolics)¹⁵, these processes create an especially recalcitrant peat that resists further mineralization, thus enabling peat accumulation despite seasonal semi-aerobic conditions down to 30–90 cm¹⁵.

Reviewer #2 (Remarks to the Author):

This paper covers a very interesting area, and involves a considerable amount of field work and analysis, but I have some fundamental issues with it.

I do not find the paper easily readable and feel that it is overly complicated in style, but with important factors missing from the discussion. In addition, I have concerns about some of the methods used and their influence on the resulting conclusions. There are a lot of aspects which I wish to discuss, but may not be able to cover everything.

Response: We thank the reviewer for their constructive comments. We have made several revisions to the text which we hope improve the clarity, including further justification for our methods (li. 524-529), an introduction of our hypothesized reasons for the presence of peat in warm climates (li. 160-170), more explicit explanation of the effects of water content (li. 220-239), and several other edits. Please see our more detailed responses below.

The key points relate to:

- Absence of values for or discussion of aliphatic or “waxy” components, which are a key feature of peat and tend to increase down the profile, and will play a significant part in how decomposable the peat is (Leifeld et al. 2012)

Response: We did not observe any systematic trends in aliphatics with latitude. We have added a supplementary figure (Supp. Fig. 3) that illustrates this, and addressed this issue in the text as follows (li. 119-121):

'Aliphatic content was slightly higher in temperate to tropical sites, but this difference was much less pronounced (Supplementary Fig. 3).'

- No explanation as to why the direct regression of the FTIR spectra against wet chemistry analysis was not used, or reasons for using the more complex method developed here using R

Response: Further justification of our analysis of pre-selected peaks has been added to the Methods (li. 524-529; text copied below):

'Since our study was focused on carbohydrate and aromatic contents of peats, we chose to isolate specific FTIR bands representative of those two functional groups and quantitate based on the peak heights of those bands. We recognize that multivariate analyses of entire FTIR spectra (such as partial least squares) can identify additional spectral features related to humification⁸⁰, but here we are quantifying carbohydrate and aromatic content using FTIR spectra correlated with wet chemical methods that only provide data on these two functional groups.'

Although direct regression of the entire spectra against the wet chemistry analyses might yield additional insights, these are beyond the scope of this study, which instead refines previous methods based on ratios of peaks (humification indices) used by our group and many others (Schultz et al., 1985; Rodrigues et al., 1998; Pandey and Pitman, 2004; Tfaily et al., 2014; Hodgkins et al., 2014). In many ways, the preselection of individual peaks is a simpler method, as it does not result in the selection of peaks that do not directly result from (but merely co-occur) with the analytes measured by wet chemistry. Our R script simply improves the precision of this peak selection by accounting for slight differences in peak location between samples. The individual peak heights, normalized to the area of the spectrum, were then regressed against the wet chemistry analyses. Please see our additional comments below for further justification of our methods.

- No explanation as to how the estimated errors for the composition of plants at each site can be presented with errors often as small as those for the peats (Fig. 3) given the arbitrary nature of deciding which plants and in which proportions may have contributed to peat formation (and the additional uncertainty about changes in climate and plant populations over time).

Response: We have changed the error bars of Fig. 3 from 95% confidence intervals (which depend on the number of samples) to standard deviations (which more closely reflect the measured variability), and added text to the figure caption (li. 245-247 copied below) that acknowledges that these errors do not account for unknowns in vegetation composition:

'Error bars represent standard deviations of the measured samples, and do not account for uncertainty in species composition of peat-forming plants.'

In regards to these vegetation unknowns, peatlands are surprisingly similar within the various climatic zones of this latitudinal gradient in terms of their plant types and major peat formers, with Sphagnum moss and sedges dominating the vegetation in boreal and sub-arctic bogs and fens (Glaser et al., 1981; Gorham, 1991), Cladium and other sedges dominating non-forested subtropical peatlands (Richardson et al., 1999; Sjögersten et al., 2011; Glaser et al., 2013), and trees, woody vines, and saplings dominating lowland tropical peat domes in SE Asia, Africa, and South America (Page et al., 2011; Dargie et al., 2017; Dommain et al., 2015; Richardson, 2012; Sjögersten et al., 2011). The published record supporting these generalizations are voluminous. The text has been clarified to reflect the ubiquity of these plant communities in different latitudinal zones (li. 172-175 copied below; emphasis added):

'The source vegetation responsible for peat formation varies with latitude, with non-woody Sphagnum and sedges dominant within a broad range of colder climates^{29,47}, and woody trees and shrubs (or less commonly Cladium and other sedges if non-forested) dominant within a broad range of warmer climates^{2,4,10,48-50}.'

- Not using the FTIR dataset to its full potential, and missing out on important information which could be obtained by further interpretation and comparison of the spectra

Response: Please see our more detailed comments below. We have added several figures to the SI (Supp. Figs. 5–8), which show the full spectra of representative subsets of samples.

FTIR analysis

The FTIR spectra could have been directly regressed against the wet chemistry results to predict the carbohydrate and aromatic content. This would have been a simpler approach and does not rely on pre-selecting which peaks to use. It seems to me that the R script for location of the peaks is unnecessary and may lead to poorer results than the direct regression (which allows the peaks which describe the observed variance to be selected). At the very least, a direct regression should have been done as a comparison to their approach to demonstrate that there is something to be gained from it.

Response: Peak analysis is the standard approach for interpreting spectral data, and this paper is a follow-up of our previous studies which have used this technique (Tfaily et al., 2014; Hodgkins et al., 2014). The reviewer is certainly correct in noting that much information can be obtained by linear regressions of entire IR spectra and subsequent principal component analyses. However, the focus of this study is the specific role of carbohydrates and aromatics, two components that have been identified as important in peat humification. Furthermore, we developed quantitative correlations with IR peaks associated with these functional groups and wet chemical analyses that allowed us to determine ABSOLUTE contents, not just relative abundances. Direct regression can certainly identify additional peaks that correlate with compound classes measured by wet chemistry; however, some of these peaks may not be directly produced by the compounds of interest, but may instead arise from other compounds that commonly co-vary with the compounds of interest. We thus chose to use the

conventional approach of isolating and directly quantifying the IR peaks of interest—specifically, peaks already well-known to arise from carbohydrates and aromatics—rather than more sophisticated statistical methods that might yield additional insights but which are beyond the scope of this study.

The calibration dataset ought to be appropriate for the population to be analysed to ensure a robust calibration and in this case it does not appear so to me, particularly for the peat analysis. I would like to see evidence that the spectra of the unknown samples being analysed fall within the spectral range of the calibration set. I also find it slightly bizarre that you would choose to use magazines for this.

Response: The standards used were chosen to represent a very broad range of carbohydrate and aromatic concentrations. A figure legend has been added to the regression equations (Supp. Fig. 2) to show how each subset of standards falls along the regression lines. Average spectra of standards used for the calibration are included in a new figure in SI (Supp. Fig. 8), along with some representative samples (Supp. Figs. 5–7). Spectra of standards clearly fall within the spectral window of samples, particularly with regards to the target peaks. The old magazine (OMG) and office paper (OFF) standards were chosen because they have very low lignin contents and are dominated by cellulose, thus allowing us to compare a wide range materials of differing composition. Despite some unusual spectral features, these standards' peak heights and wet chemistry results fall along the regression lines with the other standards (Supp. Fig. 2; see li. 556-559 copied below):

'Despite these unusual spectral features in the OMG and OFF standard sets, they still fit with the other standards (with spectra more similar to our peat samples; Supplementary Fig. 8) along the aromatic and carbohydrate calibration curves, respectively (Supplementary Fig. 2).'

A peak at 780 cm^{-1} could arise from amorphous silica but is more likely to arise from oxalate, particularly in plant material. There is a doublet at 779 and 797 cm^{-1} which is due to the presence of quartz, but there would not be a peak at this frequency for clay minerals, which are very likely to occur. These should have been taken into account by looking for peaks in the OH stretching region (e.g. 3695 and 3620 cm^{-1} for kaolinite).

Response: We have samples from the mineral layer of several of our peatlands, which were not included in this paper, but a few spectra are shown below. Note that in the Stordalen samples, the amorphous silica peak at 780 cm^{-1} is the only mineral peak observed, while the kaolinite peaks at 3695 and 3620 cm^{-1} are absent or barely visible. We thus used the 780 peak as a proxy for mineral content, as it works for all of our study sites. None of the samples included in this paper had peaks at 3695 and 3620 cm^{-1} ; nonetheless, since these peaks appeared in some deeper mineral peat layers (e.g. DNL deep and MDM11-2A core sections shown below), we have added identification of these peaks to the relevant section of the Methods (see li. 542-543):

'To avoid interference from silicates, which produce a large FTIR peak that interferes with the carbohydrate peak ($\sim 1030 \text{ cm}^{-1}$)⁸⁴, core depths that contained silicates (determined based on the presence of peaks at 3695 and 3620 cm^{-1} [kaolinite] and/or 780 cm^{-1} [amorphous silica], combined with a large peak at $\sim 1030 \text{ cm}^{-1}$) were excluded from our analysis.'

The FTIR spectra give the overall chemistry of the peat samples and I think a lot more could have been made of interpretation of the spectra. Where differences or similarities are described I would expect to see some illustrations. I am not comfortable with papers, using FTIR analysis, where spectra aren't shown. A lot of relationships could have been explored by looking at the spectra in more detail: For example a comparison between the peat samples from different latitudes with similar carbohydrate and aromatic analysis results could have shown whether there were other chemical differences apparent. Also, a look at the progressive changes in the peat spectra as you went down a profile could have illustrated whether the trends reported in Fig. 1 are clearly visible in the spectra. A comparison between the vegetation and the peat spectra would also be informative.

Response: We have added additional text that emphasizes the narrow focus of the study on carbohydrates and aromatics that have been identified as important drivers of peat decomposition (li. 108-117 and li. 524-529 copied below):

'In this study, we have focused on two solid phase organic matter components that have been shown to drive peat decomposition: carbohydrates that are the most labile solid-phase component²⁰, and aromatics that inhibit anaerobic decomposition^{14,38}. ... While other components such as aliphatics have been shown to correlate with peat humification²⁰, these components have not been identified as "active" in the humification process⁴⁰ (unlike

carbohydrates that are preferentially lost²⁰ and aromatics that can actively inhibit decomposition^{14,38}), but most likely become concentrated as labile components degrade.’ (li. 108-117)

‘Since our study was focused on carbohydrate and aromatic contents of peats, we chose to isolate specific FTIR bands representative of those two functional groups and quantitate based on the peak heights of those bands. We recognize that multivariate analyses of entire FTIR spectra (such as partial least squares) can identify additional spectral features related to humification⁸⁰, but here we are quantifying carbohydrate and aromatic content using FTIR spectra correlated with wet chemical methods that only provide data on these two functional groups.’ (li. 524-529)

We don't deny that additional factors/components might be involved, but carbohydrates and aromatics are the focus here, and we believe the data and interpretations presented here are persuasive that they are indeed reflective of peat decomposition potential. Spectra of some peat samples and plant material are included in new SI figures (Supp. Figs. 5–7), and these spectra demonstrate that the trends we quantify are indeed highly visible from the raw spectra.

I am also concerned that only four scans were averaged per sample as this is far fewer than would be recommended and risks producing spectra with poor signal to noise ratios.

Response: Spectra in Supp. Fig. 1 and in Supp. Figs. 5–8 demonstrate that four scans do provide sufficient S/N ratios for our purposes.

Peat chemistry and relationship to degradation

A large part of the change observed down a peat profile, in my experience, is the development of strong waxy type profile from long chain aliphatics – this is absent in this paper beyond showing aliphatic peaks in the R script. This appears to be a major omission, with no apparent justification. Papers by Leifeld indicate that the O-C % is important, rather than just the ratio between carbohydrates and aromatics (Leifeld et al. Geophysical Research Letters, Vol 39, L14704, 2012; Bader et al. Biogeosciences Discuss 2017)

Response: We have added text (li. 108-117) that acknowledges additional features of IR spectra can be representative of peat decomposition (e.g. aliphatics). However, there is substantially more literature suggesting that carbohydrates and aromatics are ACTIVE components in peat decomposition and aliphatics are merely concentrated due to loss of more labile components. Indeed, that seems to be the primary conclusion of Leifeld et al. (2012) and Bader et al. (2017). In particular, Fig. 2 of Bader et al. (2017) (and references cited by that paper) shows decreases in both O/C and H/C ratios with depth. This is consistent with loss of carbohydrates and accumulation of aromatics, whereas large accumulations of aliphatics would not cause a decrease in H/C. Moreover, because we did not observe strong latitudinal trends in aliphatic content (li. 119-121; Supp. Fig. 3), discussions of changes in

aliphatics down the peat profile (although important at site-level) are beyond the scope of this global analysis.

The relevant text in the revised manuscript is copied below:

'In this study, we have focused on two solid phase organic matter components that have been shown to drive peat decomposition: carbohydrates that are the most labile solid-phase component²⁰, and aromatics that inhibit anaerobic decomposition^{14,38}. ... While other components such as aliphatics have been shown to correlate with peat humification²⁰, these components have not been identified as "active" in the humification process⁴⁰ (unlike carbohydrates that are preferentially lost²⁰ and aromatics that can actively inhibit decomposition^{14,38}), but most likely become concentrated as labile components degrade.' (li. 108-117)

'Aliphatic content was slightly higher in temperate to tropical sites, but this difference was much less pronounced (Supplementary Fig. 3).' (li. 119-121)

There does not appear to be much attention paid to possibilities of changes in vegetation and rates of degradation with time and climatic conditions – beyond considering the water table. It is possible to come across pockets with high % carbohydrate buried deep in the peat, given the right circumstances. Analysis of macrofossils can provide information about changes in the vegetation which contribute to peat formation over time. (D. Mauquoy and D. Yeloff, *Biodivers Conserv* (2008) 17:2139–2151)

Response: Variability in vegetation downcore does not appear to affect our overall findings, because regressions with latitude and temperature do not change substantially when the averages over the entire cores are used instead of the average over the top 50cm. We designed our methodology to focus on peat chemistry averaged across broad depth ranges. Macrofossil and detailed peat stratigraphy studies of these sites have been already published elsewhere (e.g.: Glaser et al., 1990; Talbot et al., 2010; Kokfelt et al., 2010; Glaser et al., 2013; Bernhardt et al., 2013; Dommain et al., 2015).

Age of the peat – for Pocosin it appears that there is no new peat and so comparisons with the top of other peatlands probably aren't relevant and may explain results for this site having values lying off the lines in the diagrams in Fig. 2. I question whether this is a valid comparison as in Leifeld et al. 2012 they suggest that a peat of ~4000 years age may have originally been buried at a depth of 2m, and this would make sense because it wasn't created recently.

Response: We have added text that acknowledges the effect of the old Pocosin peat age on its high degree of transformation (li. 232-233) and appearance as an outlier in the regressions (li. 261-264):

'This high degree of transformation is consistent with the unusually old age of the peat (Fig. 4)' (li. 232-233)

'Consistent with this greater age, the NC Pocosin peat also appeared more chemically transformed compared to the other sites, as reflected in the unusually low peat carbohydrate

content and appearance as an outlier in the regressions with latitude and temperature (Fig. 2).’ (li. 261-264)

Greater changes would be expected at higher temps in the low latitudes initially which would lead to the degraded more recalcitrant peat – I would like to see how these spectra compared to lower depth high latitude peat.

Response: This is one of our main hypothesized explanations for the greater recalcitrance of low-latitude peat, which we have clarified in the text (see mechanism #2 in the copied text below, which is expanded upon in li. 200-219). We have added a new supplementary figure that illustrates that the FTIR spectra of shallow low-latitude peat are indeed similar to those of deep high-latitude peat (Supp. Fig.5; see also li. 160-170 copied below, with emphasis added):

‘The trend in peat chemistry with latitude—specifically, the lower carbohydrate and higher aromatic content in tropical and subtropical peatlands—are most pronounced at the surface, whereas northern peat at deeper depths acquires a chemistry more similar to low-latitude peat (Fig. 1; Supplementary Fig. 5). This pattern suggests two possible mechanisms to explain the global trend in peatland organic matter chemistry: (1) the initial chemical quality of the peat-forming plant material (carbohydrate and aromatic content) is changing along the latitudinal transect, such that plant litter and the resulting peat are more recalcitrant at low latitudes, or (2) there is a direct temperature control on the initial rate of labile carbon loss in peatlands, such that surface (sub)tropical peat is already well-decomposed, whereas surface northern peat is poorly decomposed and instead degrades more slowly down the profile. In addition, a combination of both mechanisms may have a role in creating this latitudinal trend in peat chemistry.’

In Fig. 3 the composition of plants at each site is presented with errors often as small as those for the peats which I find difficult to accept, given the arbitrary nature of deciding which plants and in which proportions may have contributed to peat formation. This is without the additional uncertainty introduced by changes in climate and plant populations over time. We already know that at the Pocosin site, the peat was created thousands of years ago. I remain to be convinced that the values for the vegetation of the % carbohydrate and % aromatic content can be quoted this accurately and I would like to see how the values were arrived in order to determine how valid a comparison is it with the peat.

Response: We have changed the error bars of Fig. 3 from 95% confidence intervals (which depend on the number of samples) to standard deviations (which more closely reflect the measured variability), and added text to the figure caption (li. 245-247 copied below) that acknowledges that these errors do not account for unknowns in vegetation composition:

‘Error bars represent standard deviations of the measured samples, and do not account for uncertainty in species composition of peat-forming plants.’

In regards to these vegetation unknowns, peatlands are surprisingly similar within the various climatic zones of this latitudinal gradient in terms of their plant types and major peat formers, with Sphagnum

moss and sedges dominating the vegetation in boreal and sub-arctic bogs and fens (Glaser et al., 1981; Gorham, 1991), Cladium and other sedges dominating non-forested subtropical peatlands (Richardson et al., 1999; Sjögersten et al., 2011; Glaser et al., 2013), and trees, woody vines, and saplings dominating lowland tropical peat domes in SE Asia, Africa, and South America (Page et al., 2011; Dargie et al., 2017; Dommain et al., 2015; Richardson, 2012; Sjögersten et al., 2011). The published record supporting these generalizations are voluminous. The text has been clarified to reflect the ubiquity of these plant communities in different latitudinal zones (li. 172-175 copied below; emphasis added):

'The source vegetation responsible for peat formation varies with latitude, with non-woody Sphagnum and sedges dominant within a broad range of colder climates^{29,47}, and woody trees and shrubs (or less commonly Cladium and other sedges if non-forested) dominant within a broad range of warmer climates^{2,4,10,48-50}.'

The number of sites and latitudes represented is not large and there are not multiple sites at given latitude and so I would like to have evidence for the representativeness of the sites and the justification for making to make generalisations about latitude.

Response: Based on our field experience, we believe that the coring sites provide an initial representative sample of the peat chemistry along our transect from the subarctic to the equatorial lowlands of the tropics. Please also note that this sample set contains sites from the largest peatlands in North America (RL-II Bog and RL-II Fen in the Glacial Lake Agassiz Peatlands, and Loxahatchee in the Everglades) and also one of the last intact peat domes in tropical SE Asia (Mendaram, Brunei). The coring sites selected were representative of the regional vegetation patterns that have been documented in numerous published studies dealing with the peatland floras, vegetation, and peat stratigraphy (see comment above; also see Glaser et al., 1990; Talbot et al., 2010; Kokfelt et al., 2010; Glaser et al., 2013; Bernhardt et al., 2013; Dommain et al., 2015). The general uniformity and low species diversity of high-latitude peat-forming plants (Glaser, 1992; van Breemen, 1995), and the ubiquitously high lignin and aromatic content of woody plants typically found in low-latitude peatlands (Gandois et al., 2013; Gandois et al., 2014; Wang et al., 2015), also support the representativeness of this study. Nonetheless, we have added two more cores from another site (Mer Bleue, Canada).

There appear to be some contradictions in the text in that it is stated that the southern latitudes have woody inputs and therefore they have higher aromatics but then for Loxahatchee say it is because of lignin in the roots? I would like to see a comparison of the spectra (as mentioned earlier) of the peats with the same proportions of aromatics and carbohydrates from the upper levels of southern latitudes and lower levels of northern latitudes to show that they are different and haven't just degraded quicker with the higher temperature. Correlations with temperature are shown to be slightly better than correlation with latitude (Fig. 2) so could it be the key driver?

Response: The different vegetation in forested and non-forested low-latitude sites has been clarified in the text (li. 174-175). As stated in li. 185-187, high lignin content in the Loxahatchee is derived from

bundles of sclerenchyma fibers (i.e. thick walled cells impregnated with lignin that are dead at maturity) from *Cladium jamaicense* (Metcalf, 1971). The relevant text passages are copied below:

'woody trees and shrubs (or less commonly Cladium and other sedges if non-forested) dominant within a broad range of warmer climates^{2,4,10,48-50}' (li. 174-175)

'In the Loxahatchee, despite the relative sparseness of woody plants, lignin is still abundant in the roots and shoots of Cladium jamaicense, which are strengthened by "girders" (i.e. bundles of sclerenchyma cells)⁵¹.' (li. 185-187)

In regards to temperature being the main driver of trends in peat chemistry, we have added a supplementary figure (Supp. Fig. 5) comparing selected peat samples from the upper layers of southern latitudes with lower layers of northern latitudes. While there are variations within each category, there don't appear to be systematic differences.

The low-latitude surface peat may have degraded more quickly with the higher temperature: This is, in fact, one of our two main hypothesized mechanisms for the observed latitudinal differences in surface peat. It's likely true that temperature is the key driver: But not only through faster decomposition at warmer temperatures (our 2nd hypothesis: *'there is a direct temperature control on the initial rate of labile carbon loss in peatlands' [li. 167]*), but also through differences in plant communities that grow in different climates (our 1st hypothesis: *'the initial chemical quality of the peat-forming plant material ... is changing along the latitudinal transect' [li. 164-166]*). But because plant communities are also affected by light availability, particularly in the Arctic where conditions are dark for much of the year, temperature is only one driver of these differences. We therefore describe our overall results in terms of latitudinal trends, as latitude is the driving variable of both temperature and light availability, in addition to other climate characteristics such as precipitation.

Reviewer #3 (Remarks to the Author):

This is a very interesting and novel manuscript that compares the composition of boreal peat bogs with those at low latitudes (as well as surface peat vs deeper peat). The results have potential implications for peat loss in high-latitude peatlands and thus for modelling future carbon emissions. Therefore they could be of relevance for investigations of future climate change.

Response: We thank the reviewer for their constructive comments.

You are mainly using space (latitude) to make your points. Given the large amount of palaeo work done on peat bogs, could the time component be discussed more? E.g., see Charman et al., 2013. Climate-related changes in peatland carbon accumulation during the last millennium. *Biogeosciences* 10, 929-944

Response: The latitudinal temperature gradient from the poles to the equator has existed throughout the Holocene (past 11,000 years). Moreover, temperature changes over the Holocene are small compared to the latitudinal gradient in temperatures, so this would have little effect on the conclusions of this paper. This topic is highly interesting but really should be addressed in a further

paper, as additional information on the chemical stratigraphy of tropical peats becomes available at the level of chemical resolution described for high-latitude peat.

My main problem with the manuscript is its sample size - only five regions (most with multiple sites) are reported. Can the results from this small dataset really be extrapolated and generalised? Can the vegetation chemistry really be concluded to be that constant over similar latitudes? Such questions should at least be discussed in the text. This study should perhaps be seen as a valid first step to a wider analysis of the found patterns.

Response: This analysis is reasonable given the high degree of regional similarity in boreal and subarctic peatlands, particularly in terms of the low species diversity of their major peat-forming vegetation (Glaser, 1992; van Breemen, 1995) and the similarity in lignin content between plants of the same functional type across different high-latitude subregions (see figure below). Cladium and other Cyperaceae are typical dominants of sub-tropical peatlands that are not forested (Richardson et al., 1999; Sjögersten et al., 2011; Glaser et al., 2013). Despite the high species diversity of forested tropical peat domes, the chemistry of their woody tissue is considerably less complex in terms of their high lignin and aromatic content (Gandois et al., 2013; Gandois et al., 2014; Wang et al., 2015), and so the few sample sites are reasonably representative.

The ubiquity of the studied peat-forming plant community types in different latitudinal zones has been clarified in the text (see li. 172-175 copied below; emphasis added). We have also added another site (Mer Bleue).

'The source vegetation responsible for peat formation varies with latitude, with non-woody Sphagnum and sedges dominant within a broad range of colder climates^{29,47}, and woody trees and shrubs (or less commonly Cladium and other sedges if non-forested) dominant within a broad range of warmer climates^{2,4,10,48-50}.

FIGURE: Lignin content (average \pm SE, labeled with n), estimated using the extraction method of Poorter and Villar (1997), in plant leaves from subarctic (northern Sweden), cold temperate (southern Sweden), and warm temperate (the Netherlands and Belgium) peatlands. “Moss” includes Sphagnum spp., “herb” includes graminoids and forbs, and “shrub/tree” includes woody evergreen and deciduous shrubs. Data are from Dorrepaal (2005) and Aerts et al. (2006).

How thick are the studied deposits? Were all cores sampled until reaching the bog's bottom? If not, would you expect any found trends to have continued further down?

Response: We recovered peat cores from the peat surface down to the basal peat/mineral contact.

Fig. 1: this spaghetti-diagram is very hard to read. Consider separating into more panels, e.g. a panel per site showing both %carbohydrates and %aromatics.

Response: We believe that showing all the cores on the same plot, separated by carbohydrates and aromatics, does a better job of showing the latitudinal trends. To make these trends more easily visible, we have added panels to the original figure (Fig. 1 C and D) that show the overall trends within high and low latitudes (demarcated by 45°N) using LOESS smooth curves. In addition, we have added an SI figure (Supp. Fig. 4) that shows the Fig. 1 profiles separated by site category, showing both %carbohydrates and %aromatics on each plot.

We need a map. This map could show the studied regions as well as where peatlands of different types can be found.

Response: We believe that Table 1 (formerly in the SI) is sufficient for providing information on the location of the sites. We have moved it to the main text so that the latitudes and other site characteristics are more easily visible.

Line 151, what family does C. stand for here?

Response: Cladium. This has been corrected in the revised manuscript (li. 186) and in Table 1.

Line 426, do you mean that no age-reversals occurred, or that some ages were ignored to avoid age reversals?

Response: This sentence has been clarified as follows: '*Depths with age reversals were ignored in the age-depth models*' (li. 485).

Rows 11-19 in the 14C xlsx file - Why do these values have so many decimals? Is this because you estimated ages from a plot?

Response: The ages were estimated from a plot (Fig. 7 in Tfaily et al., 2014). The extra decimal places have been hidden in the revised version.

Sufficient detail is provided in the methods to enable replication of this study.

Response: We thank the reviewer for this feedback.

References

Aerts R., van Logtestijn R. S. P. and Karlsson P. S. (2006) Nitrogen supply differentially affects litter decomposition rates and nitrogen dynamics of sub-arctic bog species. *Oecologia* **146**, 652–658.

- Bernhardt C. E., Brandt L. A., Landacre B., Marot M. E. and Willard D. A. (2013) Reconstructing vegetation response to altered hydrology and its use for restoration, Arthur R. Marshall Loxahatchee National Wildlife Refuge, Florida. *Wetlands* **33**, 1139–1149.
- van Breemen N. (1995) How *Sphagnum* bogs down other plants. *Trends Ecol. Evol.* **10**, 270–275.
- Dargie G. C., Lewis S. L., Lawson I. T., Mitchard E. T. A., Page S. E., Bocko Y. E. and Ifo S. A. (2017) Age, extent and carbon storage of the central Congo Basin peatland complex. *Nature* **542**, 86–90.
- Dommain R., Cobb A. R., Joosten H., Glaser P. H., Chua A. F. L., Gandois L., Kai F.-M., Noren A., Salim K. A., Su'ut N. S. H. and Harvey C. F. (2015) Forest dynamics and tip-up pools drive pulses of high carbon accumulation rates in a tropical peat dome in Borneo (Southeast Asia). *J. Geophys. Res. Biogeosciences* **120**, 617–640.
- Dorrepaal E. (2005) Plant growth-form and climate controls on production and decomposition in northern peatlands. Dissertation, Vrije Universiteit Amsterdam. Available at: <https://research.vu.nl/en/publications/plant-growth-form-and-climate-controls-on-production-and-decompos> [Accessed November 25, 2017].
- Gandois L., Cobb A. R., Hei I. C., Lim L. B. L., Salim K. A. and Harvey C. F. (2013) Impact of deforestation on solid and dissolved organic matter characteristics of tropical peat forests: implications for carbon release. *Biogeochemistry* **114**, 183–199.
- Gandois L., Teisserenc R., Cobb A. R., Chieng H. I., Lim L. B. L., Kamariah A. S., Hoyt A. and Harvey C. F. (2014) Origin, composition, and transformation of dissolved organic matter in tropical peatlands. *Geochim. Cosmochim. Acta* **137**, 35–47.
- Glaser P. H. (1992) Raised bogs in eastern North America – Regional controls for species richness and floristic assemblages. *J. Ecol.* **80**, 535–554.
- Glaser P. H., Hansen B. C. S., Donovan J. J., Givnish T. J., Stricker C. A. and Volin J. C. (2013) Holocene dynamics of the Florida Everglades with respect to climate, dustfall, and tropical storms. *Proc. Natl. Acad. Sci.* **110**, 17211–17216.
- Glaser P. H., Janssens J. A. and Siegel D. I. (1990) The response of vegetation to chemical and hydrological gradients in the Lost River Peatland, northern Minnesota. *J. Ecol.* **78**, 1021–1048.
- Glaser P. H., Wheeler G. A., Gorham E. and Wright H. E. (1981) The patterned mires of the Red Lake Peatland, northern Minnesota: Vegetation, water chemistry and landforms. *J. Ecol.* **69**, 575–599.
- Gorham E. (1991) Northern peatlands: Role in the carbon cycle and probable responses to climatic warming. *Ecol. Appl.* **1**, 182–195.
- Hodgkins S. B., Tfaily M. M., McCalley C. K., Logan T. A., Crill P. M., Saleska S. R., Rich V. I. and Chanton J. P. (2014) Changes in peat chemistry associated with permafrost thaw increase greenhouse gas production. *Proc. Natl. Acad. Sci. U. S. A.* **111**, 5819–5824.
- Kokfelt U., Reuss N., Struyf E., Sonesson M., Rundgren M., Skog G., Rosén P. and Hammarlund D. (2010) Wetland development, permafrost history and nutrient cycling inferred from late Holocene peat and lake sediment records in subarctic Sweden. *J. Paleolimnol.* **44**, 327–342.
- Metcalf C. R. (1971) *Anatomy of the Monocotyledons: V. Cyperaceae*. ed. C. R. Metcalfe, Clarendon Press, Oxford.

- Page S. E., Rieley J. O. and Banks C. J. (2011) Global and regional importance of the tropical peatland carbon pool. *Glob. Change Biol.* **17**, 798–818.
- Pandey K. K. and Pitman A. J. (2004) Examination of the lignin content in a softwood and a hardwood decayed by a brown-rot fungus with the acetyl bromide method and Fourier transform infrared spectroscopy. *J. Polym. Sci. Part Polym. Chem.* **42**, 2340–2346.
- Poorter H. and Villar R. (1997) The fate of acquired carbon in plants: Chemical composition and construction costs. In *Plant Resource Allocation* (eds. F. A. Bazzaz and J. Grace). Academic Press, San Diego. pp. 39–72. Available at: http://www.science.poorter.eu/1997_poorter&villar_pra.pdf.
- Richardson C. J. (2012) Pocosins: Evergreen shrub bogs of the Southeast. In *Wetland Habitats of North America: Ecology and Conservation Concerns* (eds. D. P. Batzer and A. H. Baldwin). University of California Press, Berkeley, US. pp. 189–202. Available at: <http://site.ebrary.com/lib/fsulibrary/docDetail.action?docID=10587985>.
- Richardson C. J., Ferrell G. M. and Vaithyanathan P. (1999) Nutrient effects on stand structure, resorption efficiency, and secondary compounds in Everglades sawgrass. *Ecology* **80**, 2182–2192.
- Rodrigues J., Faix O. and Pereira H. (1998) Determination of lignin content of *Eucalyptus globulus* wood using FTIR spectroscopy. *Holzforschung* **52**, 46–50.
- Schultz T. P., Templeton M. C. and McGinnis G. D. (1985) Rapid determination of lignocellulose by diffuse reflectance Fourier transform infrared spectrometry. *Anal. Chem.* **57**, 2867–2869.
- Sjögersten S., Cheesman A. W., Lopez O. and Turner B. L. (2011) Biogeochemical processes along a nutrient gradient in a tropical ombrotrophic peatland. *Biogeochemistry* **104**, 147–163.
- Talbot J., Richard P. J. H., Roulet N. T. and Booth R. K. (2010) Assessing long-term hydrological and ecological responses to drainage in a raised bog using paleoecology and a hydrosequence. *J. Veg. Sci.* **21**, 143–156.
- Tfaily M. M., Cooper W. T., Kostka J. E., Chanton P. R., Schadt C. W., Hanson P. J., Iversen C. M. and Chanton J. P. (2014) Organic matter transformation in the peat column at Marcell Experimental Forest: Humification and vertical stratification. *J. Geophys. Res. Biogeosciences* **119**, 661–675.
- Wang H., Richardson C. J. and Ho M. (2015) Dual controls on carbon loss during drought in peatlands. *Nat. Clim. Change* **5**, 584–587.

Reviewers' comments:

Reviewer #1 (Remarks to the Author):

The authors have addressed every comment. While there are still holes, I agree with their assertion that this is an initial sample of peat chemistry from six regions. I believe this may be ready for publication, pending agreement from other reviewers regarding FTIR methods.

Reviewer #2 (Remarks to the Author):

I appreciate the considerable work that has been done to address the points raised and, from my point of view, the clarification and additional material provided has substantially improved the paper.

There are a couple of things that I would still like to comment on:

1) In relation to the standards used, I understand the justification given in relation to having a wide range of carbohydrate and aromatic concentrations. However I would like to note that, in my opinion, standards should be chosen to be representative of the materials being analysed e.g. I would also have analysed some peats using wet chemistry and had them in the set of standards, along with the relevant vegetation standards. Also, looking at the spectra of the standards in Supplementary Fig. 8, the OMG sample is dominated by kaolinite and the OFF by carbonate, and for that reason, I wouldn't have chosen them to be standards for analysis of organic samples (although I know there is also organic components present). The OFF and OCC samples appear to all lie above the calibration line in Supplementary Fig. 2.

2) The appearance of the band, present at 780 cm⁻¹ in some of the spectra, is consistent with poorly resolved quartz rather than amorphous silica.

3) I disagree with the statements relating to the use of individual peaks rather than direct regression of the entire spectra against the wet chemistry. The regression process will determine which spectral features best correlate to the variance in the measured parameters, and that is often not only a single peak. It is not a case of irrelevant data being included, or looking for additional insights (although these may be provided from the loadings). It should be noted that the spectra of lignin and cellulose are both complex, exhibiting peaks across the majority of the spectral range. I would have expected the regression method to produce more accurate predictions than the individual peaks (especially given that the peak at 1630 cm⁻¹ could potentially have contributions from carboxylate and amide functional groups). The approach used here appears valid, but is not the one that I would have chosen.

4) Related to the previous point, in lines 487 to 495 I think what is qualitative, semi-quantitative and quantitative needs to be clarified

5) In the supplementary material, the spectral diagrams have the low frequencies on the x-axis at LHS, whereas usually they are presented the other way round

6) A PCA analysis of the spectral data (particularly in relation to Supp Fig. 5) would have been interesting to see whether there was overlap between the deep peat at high latt and the surface peat at low latt . If vegetation spectra were also included in a PCA that would be a good addition to Supp Fig. 6.

(Reviewer comments in normal text; author response in **teal bold**)

Editor:

In this revised version, we urge you to pay particular attention to Reviewer 2's concerns regarding the standards used in the analyses. We also encourage you to follow their recommendation to include a PCA analysis of the spectral data.

Response: We have followed the recommendation of Reviewer 2 and have added two PCAs of the spectral data, which are overlain with vectors for environmental variables and peak-height-based chemistry (Figure 4; lines 176-188, 138-145, 193-196, 251-256, and 683-691). These include (Fig. 4ab) a PCA of the peat samples, and (Fig. 4cd) a PCA of the peat + vegetation samples. We have also followed the recommendation of Reviewer 3 from the previous round of reviews, and added a map of our sites overlaid onto global mean annual temperature (Figure 1, lines 108-114).

Regarding the standards, we recognize that the calibrations would have been more robust with the inclusion of standards more like the samples. However, the calibration standards reported in Supp. Fig. 2 were samples of opportunity. A colleague had mentioned performing the time consuming and expensive carbohydrate and Klason lignin analysis on a wide range of organic materials for a completely separate study. We were extremely fortunate to be able to request and obtain subsamples of these materials for FTIR analysis, and that they shared with us their compositional data that had been obtained by "wet chemistry" analysis. The calibrated samples span a wide range of both carbohydrate and Klason lignin, and the peak heights we have associated with these parameters respond to the measured values in a linear fashion (lines 621-624 and 641-646).

The divergence of the standards from the peat samples, noted by Reviewer 2, makes them better suited for calibration using individual peak heights rather than direct regression. Because direct regression takes into account the whole spectra, it is more sensitive to the presence of compounds that differ between sample and standard sets (lines 644-646). We tried a partial least squares regression of the FTIR spectra against the wet chemistry, but the resulting regression model produced negative values for % carbohydrates in some of the peat samples, likely due to influences from other parts of the spectra. In contrast, the FTIR results for the peat samples (and their derived estimates of % carbohydrates and % aromatics) overlapped much better with the calibration standards when we used isolated peak heights. For this reason, we think that the best approach is to retain the peak height analysis. This method is widely established in the literature, as we have emphasized in the revised manuscript (lines 538-540, 545-546, and 548-549; refs. 19,21,22,77–80,84). Thus while we have deviated from the procedures followed in the laboratory of this reviewer, we believe we have followed rigorous and logical scientific procedure.

We hope that any remaining concerns about our use of preselected peaks are addressed by the PCAs we have added (Figure 4), which show that PC1 (the direction of latitudinal variation) has the strongest loadings in the peaks we used to perform the calibrations (lines 140-142).

Reviewer #1 (Remarks to the Author):

The authors have addressed every comment. While there are still holes, I agree with their assertion that this is an initial sample of peat chemistry from six regions. I believe this may be ready for publication, pending agreement from other reviewers regarding FTIR methods.

Response: We thank the reviewer for their comments.

Reviewer #2 (Remarks to the Author):

I appreciate the considerable work that has been done to address the points raised and, from my point of view, the clarification and additional material provided has substantially improved the paper. There are a couple of things that I would still like to comment on:

Response: We appreciate the reviewer's comments. We performed a lot of significant additions to the paper in the first revision in the hope of addressing the previous points raised, and hope that our further revisions and responses below address any remaining concerns. We are deeply appreciative of the reviewer's interest in our paper and considered carefully their every comment and suggestion.

1) In relation to the standards used, I understand the justification given in relation to having a wide range of carbohydrate and aromatic concentrations. However I would like to note that, in my opinion, standards should be chosen to be representative of the materials being analysed e.g. I would also have analysed some peats using wet chemistry and had them in the set of standards, along with the relevant vegetation standards. Also, looking at the spectra of the standards in Supplementary Fig. 8, the OMG sample is dominated by kaolinite and the OFF by carbonate, and for that reason, I wouldn't have chosen them to be standards for analysis of organic samples (although I know there is also organic components present). The OFF and OCC samples appear to all lie above the calibration line in Supplementary Fig. 2.

Response: We state in the text that the OMG and OFF samples were removed from calibrations of peaks that had interference from other components (lines 616-624). In the calibrations in which these samples were retained, the peaks produced by the inorganic fractions do not overlap with the carbohydrate peak (~1030) in OFF or the aromatic peaks (~1510 and ~1630 cm^{-1}) in OMG, and these samples' distances from the calibration lines (and those of OCC in the carbohydrate calibration) are comparable to those of the completely organic and natural samples (HW, SW, LG, and GN) (Supp. Fig. 2). The comparable residuals in these regressions across a broad range of samples, despite the differences in other portions of their spectra, strengthens our case for interpreting the normalized peak heights as relative abundances which are directly proportional to wet chemistry (lines 621-624, 641-646, and 651-658).

We recognize that the calibrations would have been more robust with the inclusion of standards more like the samples. However, the calibration standards reported in Supp. Fig. 2 were samples of opportunity. A colleague had mentioned performing the time consuming and expensive carbohydrate and Klason lignin analysis on a wide set of organic materials for a completely separate

study. We were extremely fortunate to be able to request and obtain subsamples of these materials for FTIR analysis, and that they shared with us their compositional data that had been obtained by “wet chemistry” analysis. The calibrated samples span a wide range of both carbohydrate and Klason lignin, and the peak heights we have associated with these parameters respond to the measured values in a linear fashion (lines 621-624 and 641-646).

Nonetheless, the reviewer raises a good point that other components in the spectra can complicate calibrations of FTIR with wet chemistry. We have therefore added new text (lines 646-650) that cautions future studies to avoid this interpretation for samples with components that overlap with the peaks of interest: “However, our approach should only be used if there are no compounds that produce large peaks that overlap with the specific peaks being analyzed (e.g., silicates that overlap with the carbohydrate peak), which there were not in this study (OMG was omitted from the carbohydrate calibration and OFF was omitted from the aromatic calibration).”

2) The appearance of the band, present at 780 cm^{-1} in some of the spectra, is consistent with poorly resolved quartz rather than amorphous silica.

Response: We have changed “amorphous silica” to “poorly resolved silicate minerals” (lines 609-610). This wording reflects that we’ve observed this peak in several silicates: not just quartz, but also kaolinite and montmorillonite.

3) I disagree with the statements relating to the use of individual peaks rather than direct regression of the entire spectra against the wet chemistry. The regression process will determine which spectral features best correlate to the variance in the measured parameters, and that is often not only a single peak. It is not a case of irrelevant data being included, or looking for additional insights (although these may be provided from the loadings). It should be noted that the spectra of lignin and cellulose are both complex, exhibiting peaks across the majority of the spectral range. I would have expected the regression method to produce more accurate predictions than the individual peaks (especially given that the peak at 1630 cm^{-1} could potentially have contributions from carboxylate and amide functional groups). The approach used here appears valid, but is not the one that I would have chosen.

Response: The aim of our approach was not just to estimate percentages of cellulose and lignin, but also to evaluate the interpretation of area-normalized peak heights, more generally, as directly correlated to the weight percentages of their corresponding compounds. Our use of individual peaks was designed to directly accomplish this aim, which we have clarified in the revised manuscript (lines 557-560).

The reviewer raises a good point that direct regression can determine which spectral features best correlate with wet chemistry, and that these might differ from pre-selected peaks. We hope these concerns are addressed by the PCAs we have added (Figure 4), which show that PC1 (the direction of latitudinal variation) has the strongest loadings in the peaks that we used to perform the calibrations. This result has been described in the text as follows: “In both PCAs, the loadings of PC1 were most negative in the peak at $\sim 1030 \text{ cm}^{-1}$ (used to estimate % carbohydrates) and most positive in the peaks at ~ 1500 and $\sim 1600 \text{ cm}^{-1}$ (used to estimate % aromatics) (Fig. 4ac)” (lines 140-142).

The use of individual pre-selected peaks is also widely established in the literature, as we have emphasized in the revised manuscript (lines 538-540, 545-546, and 548-549; refs. 19,21,22,77–80,84).

For example, Tfaily et al (2014) (ref. 21) used peak heights to analyze FTIR spectra at a site (S1 Bog) we included in this study. They took the ratio of various peak heights relative to the carbohydrate peak height and calculated humification indexes. Similarly, Rodrigues (1998) and Pandey and Pitman (2003, 2004) (refs. 78-80) used baseline corrections similar to our method, and calibrated ratios of peak heights relative to lignin content measured by wet chemistry. Thus while we have deviated from the procedures followed in the laboratory of this reviewer, we believe we have followed rigorous and logical scientific procedure and followed a robust line of research well represented in the literature.

Our study followed in this established line of research, with an additional refinement: Rather than calculating ratios of peak heights, which makes it difficult to discern which compound class is driving changes in these ratios, we isolated each compound class by normalizing the peak heights to the integrated area of the spectrum. This method follows on the individual peak calibrations used by da Costa Lopes et al (2013) (ref. 84). Our calibrations have shown that these normalized peak heights are directly proportional to the absolute concentrations of two compound classes (carbohydrates and aromatics), which suggests that the same may be true of other compounds. For example, although we did not measure aliphatics by wet chemistry, our study provides strong evidence that such a measurement would produce a linear correlation with the aliphatic FTIR peak (aliph29, $\sim 2920\text{ cm}^{-1}$) similar to those shown for carbohydrates and aromatics (lines 651-655). Thus, if an aliphatic calibration were performed in a future study, and the aliphatic abundances in Supp. Fig. 3 were then re-plotted using the resulting percentages, the resulting graph would look the same except for the scale and units of the x-axis.

In contrast to our regressions with individual peaks, direct regressions of the entire spectra produce results that are specific to the compounds analyzed by wet chemistry. As such, they do not provide any basis for estimating relative abundances of compounds not measured. In other words, if we had used whole spectra regressions instead of regressions of individual peak heights, this approach would not have provided any evidence to support interpreting the aliphatic relative abundances as directly proportional to aliphatic weight percentages. Although we could have used both approaches, we chose to still use peak heights for carbohydrates and aromatics, as this allows for more direct comparison between these calibrated weight percentages and non-calibrated peak relative abundances.

Most importantly, because direct regressions account for everything in the FTIR spectra, they are much more sensitive to the presence of non-analyte compounds that differ between sample and standard sets (see the reviewer's point #1 and our response above, and also lines 644-646 of the revised manuscript). We tried a partial least squares regression of the FTIR spectra against the wet chemistry, but the resulting regression model produced negative values for % carbohydrates in some of the peat samples, likely due to influences from other parts of the spectra. In contrast, the FTIR results for the peat samples (and their derived estimates of % carbohydrates and % aromatics) overlapped much better with the calibration standards when we used isolated peak heights. After considerable research and deliberation, we have thus decided to retain our original method.

4) Related to the previous point, in lines 487 to 495 I think what is qualitative, semi-quantitative and quantitative needs to be clarified

Response: We have revised these lines to clarify the quantitiveness of different FTIR analysis methods, and also to better explain the rationale for our method (lines 533-560; see also lines 655-661).

5) In the supplementary material, the spectral diagrams have the low frequencies on the x-axis at LHS, whereas usually they are presented the other way round

Response: We have reversed the direction of all the spectral axes (Supp. Figs. 1 and 5-8).

6) A PCA analysis of the spectral data (particularly in relation to Supp Fig. 5) would have been interesting to see whether there was overlap between the deep peat at high latt and the surface peat at low latt . If vegetation spectra were also included in a PCA that would be a good addition to Supp Fig. 6.

Response: We thank the reviewer for this suggestion, and have added two PCAs of the spectral data, which are overlain with vectors for environmental variables and peak-height-based chemistry (Figure 4; lines 176-188). These include (Fig. 4ab) a PCA of the peat samples, and (Fig. 4cd) a PCA of the peat and vegetation samples. These show some overlap between deep high-latitude peat (light filled symbols) and surface low-latitude peat (dark unfilled symbols), though there is also some separation along PC1. This supports our interpretation that two factors—differing plant inputs, and more extensive peat transformation at low latitudes—contribute to the changes in peat chemistry along the latitudinal transect. More detailed interpretation (lines 138-145, 193-196, and 251-256) and methods (lines 683-691) for these PCAs are provided in the text.

REVIEWERS' COMMENTS:

Reviewer #2 (Remarks to the Author):

Again I appreciate the work done to address the points raised, and do understand now more fully the reasoning behind the approach. I wish to make a few minor points but otherwise agree that it may be ready for publication

- 1) It is unfortunate that the standards available were not more appropriate to the samples being studied, but I feel that the author's justification for their use in this study is sufficient. I do, however, still believe that better calibrations would be possible using PLS if a more appropriate dataset were available.
- 2) I was very pleased to see the two PCAs of the spectral data and think that they are very informative, and a good addition to the paper.
- 3) In relation to the band at 780 cm⁻¹ it was the appearance (or shape) of the band rather than the position which looked to me as if it were a poorly resolved quartz doublet (if properly resolved seen at 797 and 779 cm⁻¹). The authors are right that clay minerals can often have a peak near here (as does calcium oxalate). I suggest just putting in "silicate minerals".
- 4) I don't disagree that there is a rigorous and logical scientific procedure behind what the authors have done, and have often used ratios of peak heights in my own work. I just felt that in this case it might have been simpler and more accurate to use PLS, although I can now see why doing that might have been an issue with the standards which were available.
- 5) I don't fully follow how the peak area method can be used to estimate the relative abundance of compounds not measured, when the height or area of peaks is not just related to the concentration of the component but also to the relative intensity of any particular infrared absorption, which varies greatly depending on the dipole change for vibration of that functional group.

(Reviewer comments in normal text; author response in **teal bold**. Line numbers refer to the merged PDF version with tracked changes hidden.)

Reviewer #2 (Remarks to the Author):

Again I appreciate the work done to address the points raised, and do understand now more fully the reasoning behind the approach. I wish to make a few minor points but otherwise agree that it may be ready for publication

Response: We thank the reviewer for their substantial work in helping us to improve this paper.

1) It is unfortunate that the standards available were not more appropriate to the samples being studied, but I feel that the author's justification for their use in this study is sufficient. I do, however, still believe that better calibrations would be possible using PLS if a more appropriate dataset were available.

Response: We thank the reviewer for accepting this limitation. We agree that it would have been more ideal to have a standard set more like the samples, which would have allowed PLS regression.

2) I was very pleased to see the two PCAs of the spectral data and think that they are very informative, and a good addition to the paper.

Response: We agree, and thank the reviewer for this great suggestion.

3) In relation to the band at 780 cm⁻¹ it was the appearance (or shape) of the band rather than the position which looked to me as if it were a poorly resolved quartz doublet (if properly resolved seen at 797 and 779 cm⁻¹). The authors are right that clay minerals can often have a peak near here (as does calcium oxalate). I suggest just putting in "silicate minerals".

Response: The phrase "poorly resolved silicate minerals" has been changed to simply "silicate minerals" (line 454).

4) I don't disagree that there is a rigorous and logical scientific procedure behind what the authors have done, and have often used ratios of peak heights in my own work. I just felt that in this case it might have been simpler and more accurate to use PLS, although I can now see why doing that might have been an issue with the standards which were available.

Response: We thank the reviewer for their constructive feedback. We agree that PLS would have been a better method, had the standards been more appropriate.

5) I don't fully follow how the peak area method can be used to estimate the relative abundance of compounds not measured, when the height or area of peaks is not just related to the concentration of the component but also to the relative intensity of any particular infrared absorption, which varies greatly depending on the dipole change for vibration of that functional group.

Response: The phrase "relative abundances" is meant to be interpreted for the same band across samples, and not for comparison of different bands within the same sample. We have clarified this in the revised manuscript.

Nonetheless, we recognize that the relative intensity of one band across samples can vary not only based on the concentration of the compound class contributing to that band, but also on finer

variations in the functional groups (with differing infrared absorption intensities) contained within that compound class. These statements in the paper have therefore been re-worded to be more speculative. As part of this, we have deleted all specific assertions that the normalized peak heights are directly proportional to the concentrations of compounds not measured.

Specific changes are quoted below (insertions in blue; deletions in ~~gray strikethrough~~):

“Thus, this study not only provides a method for estimating concentrations of carbohydrates and aromatics, it also ~~provides a basis for interpreting~~ **suggests that other area-normalized peak heights more generally may be interpretable as relative abundances for cross-sample comparison of individual compound classes**. ~~directly correlated to the weight percentages of their corresponding compounds.~~” (lines 415-418)

“More broadly, our study suggests that even for uncalibrated compound classes (such as aliphatics [Supplementary Fig. 3] and organic acids), normalization of peak heights to spectral area (as was done in this study) ~~allows the determination of~~ **may provide a means of estimating a compound’s relative abundances across samples without normalization to any other single FTIR peak (as is the case with the commonly-used humification indices**^{18,19,21,23,77–80}). ~~In other words, FTIR is not just a qualitative method for analyzing general trends in sample chemistry, but it can also be used to derive compound class relative abundances (i.e., normalized peak heights) that are directly proportional to the concentrations of their corresponding compounds.~~” (lines 505-509)